# Development of the drop Freezing Ice Nuclei Counter (FINC), intercomparison of droplet freezing techniques, and use of soluble lignin as an atmospheric ice nucleation standard

Anna J. Miller[1,*], Killian P. Brennan[2,*], Claudia Mignani[3], Jörg Wieder[2], Robert O. David[4], and Nadine Borduas-Dedekind[1,2]

[1]Institute for Biogeochemistry and Pollutant Dynamics, ETH Zurich, Zurich, 8092 Switzerland
[2]Institute for Atmosphere and Climate Science, ETH Zurich, Zurich, 8092 Switzerland
[3]Department of Environmental Sciences, University of Basel, Basel, 4056 Switzerland
[4]Department of Geosciences, University of Oslo, Oslo, 0315 Norway
[*]These authors contributed equally to this work.

**Correspondence:** Nadine Borduas-Dedekind (nadine.borduas@usys.ethz.ch, @nadineborduas)

**Abstract.** Aerosol-cloud interactions, including the ice nucleation of supercooled liquid water droplets caused by ice nucleating particles (INPs) and macromolecules (INMs), are a source of uncertainty in predicting future climate. Because of INPs' and INMs' spatial and temporal heterogeneity in source, number, and composition, predicting their concentration and distribution is a challenge, requiring apt analytical instrumentation. Here, we present the development of our drop Freezing Ice Nucleation

Counter (FINC), a droplet freezing technique (DFT), for the estimation of INP and INM concentrations in the immersion freezing mode. FINC's design builds upon previous DFTs and uses an ethanol bath to cool sample aliquots while detecting freezing using a camera. Specifically, FINC uses 288 sample wells of $5-60$ μL volume, has a limit of detection of $-25.4 \pm 0.2$ °C with 5 μL, and has an instrument temperature uncertainty of $\pm 0.5$ °C. We further conducted freezing control experiments to quantify the non-homogeneous behavior of our developed DFT, including the consideration of eight different sources of

contamination.

     As part of the validation of FINC, an intercomparison campaign was conducted using an NX-illite suspension and an ambient aerosol sample with two other drop-freezing instruments: ETH's DRoplet Ice Nuclei Counter Zurich (DRINCZ) and University of Basel's LED-based ice nucleation detection apparatus (LINDA). We also tabulated an exhaustive list of peer-reviewed DFTs, to which we added our characterized and validated FINC.

In addition, we propose herein the use of a water-soluble biopolymer, lignin, as a suitable ice nucleating standard. An ideal INM standard should be inexpensive, accessible, reproducible, unaffected by sample preparation, and consistent across techniques. First, we compared lignin's freezing temperature across different drop-freezing instruments, including on DRINCZ and LINDA, and then determined an empirical fit parameter for future drop freezing validations. Subsequently, we showed that commercial lignin has a consistent ice nucleating activity across product batches, and demonstrated that the ice nucleating

ability of aqueous lignin solutions are stable over time. With these findings, we present lignin as a good immersion freezing standard for future DFT intercomparisons in the research field of atmospheric ice nucleation.

## Contents

## 1   Introduction

Aerosol-cloud interactions are a source of uncertainty in predicting future radiative forcing (IPCC, 2013). One important
aerosol-cloud interaction is the ice nucleation of supercooled liquid water droplets caused by ice nucleating particles (INPs).
Heterogeneous freezing can occur at temperatures as warm as −1 °C for certain bacterial (e.g., *P. syringae*; Morris et al.,
2004) and fungal (e.g., *Fusarium* species; Richard et al., 1996) INPs as well as for other currently unidentified warm INPs
(Lloyd et al., 2020). INPs typically include solid surfaces such as dust and cellular material which template ice, but recently
reported ice nucleating macromolecules (INMs) are also capable of freezing supercooled water droplets (e.g., Pummer et al.,
2012, 2015; Felgitsch et al., 2018; Kunert et al., 2019). INMs are defined here as operationally dissolved organic matter
passing through a 0.2 μL filter (Borduas-Dedekind et al., 2019). In the absence of INPs and INMs, cloud droplets with an
average radius of 10 μm remain liquid until instantaneous (< 1 s) homogeneous nucleation at approximately −38 °C (Koop
and Murray, 2016; Kanji et al., 2017). The immersion freezing mode dominates heterogeneous freezing in mixed-phase clouds
(Hoose et al., 2010; de Boer et al., 2011; Murray et al., 2012; Westbrook and Illingworth, 2013; Tobo, 2016; Kanji et al.,
2017) and occurs when an INP or an INM nucleates ice from within a supercooled water droplet (Storelvmo, 2017; Vali et al.,
2015). For instance, Hoose et al. (2010) reported that more than 85% of heterogeneous freezing events in their simulation
occurred via the immersion mode. Following ice nucleation, the ice crystal concentration in mixed-phase clouds can rapidly
increase by secondary ice processes, affecting the ratio of liquid water to ice crystals. This ratio impacts cloud microphysics,
and thus the lifetime, optical density, and radiative properties of clouds, thereby impacting the hydrological cycle and climate
(Lohmann et al., 2016; Storelvmo, 2017; Zhao et al., 2019). Indeed, Heymsfield et al. (2020) recently reported that up to 77%
of global surface precipitation originates from the ice phase. Thus, the ability to predict INP and INM concentrations can
improve estimates of primary and secondary ice concentrations in mixed-phase clouds, and thus help reduce uncertainties in
weather and climate projections (Murray et al., 2021).

This prediction is challenging due to the spatial and temporal heterogeneity in source, number, and composition of INPs and INMs. In order to reduce uncertainties, advanced methods are needed to quantify and characterize INPs and INMs from ambient and laboratory samples. A variety of laboratory instruments has been developed to measure INPs in the immersion freezing mode. Methods include continuous flow diffusion chambers (e.g., Rogers, 1988), single-particle levitation apparatuses (e.g., Diehl et al., 2014), and bench-top droplet freezing techniques (DFTs) (e.g., Hill et al., 2014). Bench-top methods vary by many variables, including cooling method, droplet generation, droplet size, droplet number, freezing detection method, detectable freezing temperature ranges, and measurement uncertainties. Cooling methods typically use either a cold stage (e.g., Wright and Petters, 2013; O'Sullivan et al., 2014; Budke and Koop, 2015; Tobo, 2016; Chen et al., 2018b, a; Häusler et al., 2018; Mignani et al., 2019; Tarn et al., 2020), a block cooled with liquid refrigerant (e.g., Hill et al., 2014; Kunert et al., 2018; Steinke et al., 2020), or a liquid cooling bath (e.g., Stopelli et al., 2014; Beall et al., 2017; Chen et al., 2018b; David et al., 2019; Gute and Abbatt, 2020). Droplet generation includes micropipetting (e.g., Hill et al., 2014; Chen et al., 2018a; David et al., 2019), shaking a vial to make an emulsion (e.g., Pummer et al., 2012; Wright and Petters, 2013), piezo-driven droplet generation (e.g., Peckhaus et al., 2016), microfluidic flow-focusing droplet generation (e.g., Stan et al., 2009; Reicher et al., 2018; Brubaker et al., 2020; Tarn et al., 2020), or filled cavities on a chip (Häusler et al., 2018). Droplet sizes and numbers vary by generation method, where pipetting typically produces fewer microliter-sized drops and where microfluidic devices produce a larger number of nanoliter-sized droplets. Droplets can be placed either on plates coated in a hydrophobic substance such as petroleum jelly, or in plastic wells such as within a multi-well PCR tray. Freezing can be detected optically with manual visual inspection (e.g. Creamean et al., 2018; Hill et al., 2014), with software to detect freezing optically (e.g., Stopelli et al., 2014; David et al., 2019; Perkins et al., 2020; Gute and Abbatt, 2020), with pyroelectrics (e.g., Cook et al., 2020), or with infrared thermal detection (e.g., Zaragotas et al., 2016; Harrison et al., 2018; Kunert et al., 2018).

Each bench-top immersion freezing method has its advantages and disadvantages which vary depending on the samples of interest. Herein, we compiled a summary of multi-drop bench-top immersion freezing instruments used for atmospheric ice nucleation measurements published between 2000 and 2020 (Table 1). Included in this summary table is a brief description of the operation of each instrument, the water background with the reported protocol, the average drop size, and the average number of droplets per experiment. Generally, advantageous qualities include large operating temperature ranges, low background freezing temperatures, and high number of drops per experiment. As these types of instruments are not yet commercial, we also built our own drop Freezing Ice Nuclei Counter (FINC) using a cooling bath and an optical detection method. In comparison to the existing methods, FINC fits well within the range of operating parameters with drop sizes of 5 - 60 μL, 288 drops per experiment, and background freezing at −25 °C (Table 1). Noteworthy features of FINC compared to existing methods are its automation of the ethanol level, its use of 288 wells to increase statistics, and its improved code for well detection and for harmonizing the output data.

With an increasing number of research groups developing DFTs, there is an ongoing search for suitable standards for freezing temperature intercomparisons. A typical standard used to compare immersion freezing instruments is the mineral dust NX-illite, a known ice-active mineral and a cheap and readily available material (Hiranuma et al., 2015a). However, NX-illite measurements can differ by orders of magnitude across different instruments (Hiranuma et al., 2015a). This discrepancy may

be due to NX-illite's insolubility in water, creating a suspension rather than a homogeneous solution. In practice, NX-illite suspensions settle quickly, potentially leading to a range of freezing temperatures. Cellulose has also been used as an intercomparison standard (Hiranuma et al., 2019), as it is the most abundant biopolymer in the environment and can contribute to ice nucleation in clouds below about −21 °C (Hiranuma et al., 2015b). However, cellulose is also a suspension in water. Snomax® has additionally been used as a bacterial ice nucleating standard and consists of freeze-dried, irradiated cells from *P. syringae* (Wex et al., 2015). Unfortunately, Polen et al. (2016) found that solutions of Snomax can have irreproducible ice nucleating activity over time, making it a rather poor standard.

An alternative to mineral dust, cellulose, and Snomax is the use of a water-soluble organic material as a standard in immersion freezing experiments. Here, we show that commercial lignin, a complex organic polymer from the cell wall structure of vascular plants (Ciesielski et al., 2020), can serve as a reproducible standard for ice nucleation across different immersion freezing techniques. Indeed, lignin is a water-soluble macromolecule with an ice nucleating activity, thereby qualifying it as an INM (Pummer et al., 2015; Bogler and Borduas-Dedekind, 2020; Steinke et al., 2020). Furthermore, lignin and its oxidation products are present in the atmosphere, emitted for example during agricultural harvesting and biomass burning in Houston, Texas with typical plume concentrations of 149 ng m$^{-3}$ (Myers-Pigg et al., 2016; Shakya et al., 2011). Lignin is also produced as a by-product of the industrial kraft process, in which wood is converted to wood pulp and subsequently used for paper products (Harkin, 1969). Recent research has shown lignin to be ice active, albeit with colder freezing temperatures than leaf litter and agricultural dust (Steinke et al., 2020; Bogler and Borduas-Dedekind, 2020). Several other studies have shown that plant materials, which may have included lignin, can be ice active in immersion freezing (e.g., Conen et al., 2016; Felgitsch et al., 2018; Suski et al., 2018; Gute and Abbatt, 2020).

Herein, we present (1) the development, characterization and validation of our home-built FINC for the quantification of INP and INM concentrations in the immersion freezing mode, (2) the intercomparison of DFTs to validate FINC, and (3) the use of soluble lignin as an intercomparison standard. As part of our intercomparison study with lignin, we show results with two other drop-freezing instruments: ETH Zurich's DRoplet Ice Nuclei Counter Zurich (DRINCZ; David et al., 2019) and University of Basel's LED-based Ice Nucleation Detection Apparatus (LINDA; Stopelli et al., 2014). We conclude by recommending commercial lignin as a standard to validate DFTs based on a detailed analysis of lignin's reproducibility and stable IN activity.

Table 1: List of drop-freezing instruments, ordered by year of first publication. Information includes the instrument name, a brief description of the instrument, the freezing temperature of a water background freezing experiment according to the reported protocol, the droplet size used, the number of droplets per experiment (drops/expt), and the main references associated with the instrument, including the reference in which the instrument was first published and, if applicable, a following reference with updates. "n.f." indicates information not found.

| Instrument name | Brief description | Water background (˚C) | Drop size | Drops/expt. | References |
|---|---|---|---|---|---|
| Flow cell microscopy technique for aerosol phase transitions | Vapors condensed onto bottom of sample cell on aluminum cooling block; freezing monitored via microscope | −37 | 7 - 33 μm diam. | ~65 | Salcedo et al. (2000), Koop et al. (2000a) |
| Droplet freezing technique (DFT) | Particles deposited on glass slide in a sample cell on a cold stage, droplets grown by water vapor; freezing monitored via microscope | −37 | 120 μm diam. | <100 | Dymarska et al. (2006), Mason et al. (2015) |
| Microfluidic apparatus | Flow-focusing nozzle continuously produces droplets in a stream of fluorocarbon across a 7-temperature-zone cold plate; freezing monitored via microscope | −37 | 80 μm diam. | >10,000 | Stan et al. (2009) |
| FRankfurt Ice Deposition freezinG Experiment - Tel Aviv University (FRIDGE-TAU) | Pipetted drops onto Vaseline-coated Peltier cold stage in low-pressure diffusion chamber; freezing monitored via CCD camera | −35 | 2 μL | ~120 | Bundke et al. (2008), Ardon-Dryer et al. (2011) |
| Picoliter and Nanoliter Nucleation by Immersed Particle Instrument (pico-NIPI, nano-NIPI) | Nebulized droplets encased in silicon oil on hydrophobic glass slides on aluminum cold stage; freezing monitored via microscope | −37 | 0.25 - 1.7 pL; 0.1 - 6 nL | ~135 (pL); ~51 (nL) | Murray et al. (2010), O'Sullivan et al. (2014), |
| Vienna Optical Droplet Crystallization Analyzer (VODCA) | Water-oil emulsion pipetted onto glass slide on Peltier cold stage, all contained in air-tight cell; freezing monitored via microscope | −36 | 10 - 200 μm diam. | n.f. | Pummer et al. (2012) |
| drop freezing apparatus for filters | Filter cutouts placed inside small tubes with water, cooled in a water bath; freezing monitored by manual inspection | −12 | 0.1 mL | 108 | Conen et al. (2012) |
| Microliter Nucleation by Immersed Particle Instrument (microL-NIPI) | Drops pipetted on hydrophobic glass slide in humidity-controlled enclosure on a cold stage; freezing monitored via camera | −26 | 1 μL | 40 | Atkinson et al. (2013), Whale et al. (2015) |
| North Caroline State University cold stage (NC State-CS) | Emulsion of water in squalene placed on a glass slide resting in an aluminum dish on a thermoelectric element; freezing monitored via camera | −34 to −36 | 400 pL - 150 nL | 300 - 1500 | Wright et al. (2013), Hiranuma et al. (2015a) |
| microfluidic device for homogeneous ice nucleation | Microfluidically produced water-in-oil emulsion on cryo-microscopy cold stage; freezing monitored via microscope. (Alternatively frozen with differential scanning calorimetry) | −36 to −37 | 53 - 96 μm diam. | >1000 | Riechers et al. (2013) |

| Instrument name | Brief description | Water background (°C) | Drop size | Drops/expt. | References |
|---|---|---|---|---|---|
| LED-based Ice Nucleation Detection Apparatus (LINDA) | Sample in tubes held in polycarbonate tray atop an LED array submersed in water-glycerin cooling bath; freezing monitored via camera | −15 | 40 - 400 µL | 52 | Stopelli et al. (2014) |
| Colorado State University Ice Spectrometer (CSU-IS) | Sample aliquots pipetted into two 96-well PCR trays cooled on custom cold blocks with $N_2$ flow; freezing monitored via camera | −25 | 50 µL | 192 | Hill et al. (2014), Hiranuma et al. (2015a), Barry et al. (2021) |
| Bielefeld Ice Nucleation ARraY (BINARY) | Droplets pipetted on glass slide with separated compartments atop a Peltier cold stage, all enclosed in a $N_2$-purged chamber; freezing monitored via camera | n.f. | 0.5 - 5 µL | 36 | Budke and Koop (2015) |
| Water-Activity-Controlled Immersion Freezing Experiment (WACIFE) | Droplets pipetted onto glass plate in a humidity-controlled aerosol conditioning cell then sealed from ambient air, and cooled on a cold stage; freezing monitored via microscope | −37 | 60 - 129 µm diam. | 30 - 50 | Wilson et al. (2015) |
| National Institute of Polar Research Cryogenic Refrigerator Applied to Freezing Test (NIPR-CRAFT) | Drops pipetted onto Vaseline-coated aluminum plate, cooled on a cryogenic refrigerator stage; freezing monitored via camera | −33 | 5 µL | 49 | Tobo (2016) |
| Karlsruhe Institute of Technology Cold Stage (KIT-CS) | Droplets printed on silicon substrate by piezo-driven drop-on-demand generator, drops then covered in silicone oil and placed on cold stage; freezing monitored via CCD camera | −36 | 215 ± 70 pL | ≥1500 | Peckhaus et al. (2016) |
| Microplate partially submersed in cooling liquid | Droplets contained in 96-well microplates partially submersed in a cooling water-alcohol bath; freezing monitored via infrared camera | −17.3 | 150 µL | 96 - 768 | Zaragotas et al. (2016) |
| Carnegie Mellon University Cold Stage (CMU-CS) | Droplets of water in oil on a substrate in aluminum chamber cooled with a thermoelectric element; freezing monitored via microscope | −27 to −28 | 1 or 0.1 µL | 30-40 | Polen et al. (2016), Polen et al. (2018) |
| microfluidic device and cold stage | Microfluidically generated drops in oil on glass slide on a cryostage; freezing monitored via microscope camera | −37 | 35 µm diam. | 200 | Weng et al. (2016) |
| Automated Ice Spectrometer (AIS) | Drops in two 96-well PCR trays fitted into aluminum blocks fixed in a liquid cooling bath, all enclosed in acrylic box; freezing monitored via camera | −25 to −27 | 50 µL | 192 | Beall et al. (2017) |
| National Oceanic and Atmospheric Administration Drop Freezing Cold Plate (NOAA-DFCP) | Drops pipetted onto Vaseline-coated copper disc placed on a thermoelectric cold plate and covered in a plastic dome; freezing monitored optically | −30 | 2.5 µL | 100 | Creamean et al. (2018) |
| PeKing University Ice Nucleation Array (PKU-INA) | Drops pipetted into compartments on a glass slide atop a cold stage in a $N_2$-purged box; freezing monitored via CCD camera | −26 | 1 µL | 90 | Chen et al. (2018a) |

| Instrument name | Brief description | Water background (˚C) | Drop size | Drops/expt. | References |
|---|---|---|---|---|---|
| WeIzmann Supercooled Droplets Observation on Microarray (WISDOM) | Microfluidically-produced droplet array on PDMS surface, placed on a cryostage purged with $N_2$; freezing monitored via microscope camera | −36 | 40 or 100 µm diam. | 550 (40 µm) or 120 (100 µm) | Reicher et al. (2018) |
| Twin-plate Ice Nucleation Assay (TINA) | Droplets contained in 4 multiwell plates (2x96 and 2x384) placed on 2 custom aluminum cooling blocks; freezing monitored via infrared camera | −25 (96-well), −28 (384-well) | 3 µL | 960 | Kunert et al. (2018) |
| Freezing on a chip | Drops loaded on a silicon plate with etched cavities and set on thermoelectric cooler in a $N_2$-flushed cell; freezing monitored via camera | −37.5 | 4 - 300 pL | 25 | Häusler et al. (2018) |
| InfraRed Nucleation by Immersed Particles Instrument (IR-NIPI) | Drops pipetted into 96-well plate on a cold stage enclosed in a chamber; freezing monitored via infrared camera | −22 | 50 µL | 96 | Harrison et al. (2018) |
| Ice Nucleation Droplet Array (INDA) | Samples placed in wells of a 96-well PCR tray cooled in a cooling bath; freezing monitored via CCD camera | −25 | 50 µL | 96 | Chen et al. (2018b) |
| Leipzig Ice Nucleation Array (LINA) | Droplets pipetted into compartments on glass slide, cooled on a Peltier element; freezing monitored via CCD camera | −30 | 1 µL | 90 | Chen et al. (2018b) |
| Microfluidic droplet freezing assay | Microfluidically-produced droplets in oil collected in microwells on glass slides placed on Peltier cold stage in airtight chamber; freezing monitored via microscope camera | −35 | 83 - 99 µm diam. | 250 - 500 | Tarn et al. (2018) |
| drop freeze assay experiment directly on exposed filters | Droplets pipetted onto filters placed on a glass slide and cold stage in a $N_2$-purged chamber; freezing monitored via camera | −30 | 1 µL | ≤ 130 | Price et al. (2018) |
| West Texas Cryogenic Refrigerator Applied to Freezing Test (WT-CRAFT) | Drops pipetted onto Vaseline-coated aluminum plate, cooled on a cryogenic refrigerator stage; freezing monitored via camera | −26 | 3 µL | 49 | Hiranuma et al. (2019) |
| DRoplet Ice Nuclei Counter Zurich (DRINCZ) | Droplets pipetted into 96-well PCR tray submersed in ethanol cooling bath; freezing monitored via camera | −22.5 | 50 µL | 96 | David et al. (2019) |
| cold stage to detect the most active INP in single crystals | Ice crystals placed with ultrapure water on a copper cold plate, melted and refrozen; freezing monitored via manual inspection | −25 | 3 µL | 4 | Mignani et al. (2019) |
| "Store and create" microfluidic device | Microfluidically-generated droplets in oil in microwells of a PDMS chip placed on a cold plate sealed with acrylic lid; freezing monitored via microscope camera | −34 | 6 nL | ≤720 | Brubaker et al. (2020) |
| pyroelectic thermal sensor for ice nucleation | Drops pipetted onto Vaseline-coated pyroelectric polymer atop a cooling block; freezing monitored via pyroelectric thermal sensor | −27 | 1 µL | 30 | Cook et al. (2020) |

| Instrument name | Brief description | Water background (°C) | Drop size | Drops/expt. | References |
|---|---|---|---|---|---|
| University of Toronto Drop Freezing Technique (UT-DFT) | Drops pipetted into multi-well PCR trays cooled in an ethylene glycol water bath; freezing monitored via camera | −23 | 50 μL | 48 | Gute and Abbatt (2020) |
| Ice Nucleation SpEctrometer of the Karlsruhe Institue of Technology (INSEKT) | Drops pipetted into two 96-well PCR trays cooled in custom cooling blocks; freezing monitored via camera | −20 | 50 μL | 192 | Steinke et al. (2020) (Schiebel (2017)) |
| Lab-On-a-Chip Nucleation by Immersed Particle Instrument (LOC-NIPI) | Water-in-oil droplets mirofluidically generated in continuous flow and passed over a series of Peltier cold plates in $N_2$-purged container; freezing monitored via microscope camera | −36 | 80 - 100 μm diam. | >1000 | Tarn et al. (2020) |
| Freezing Ice Nuclei Counter (FINC) | Drops pipetted into three 96-well Piko PCR trays submersed in ethanol cooling bath; freezing monitored via camera | −25 | 5 - 60 μL | 288 | this work |

## 2 Instrument development

### 2.1 Building components

#### 2.1.1 Hardware

The hardware design of FINC is based on DRINCZ (David et al., 2019), on its predecessors (Stopelli et al., 2014; Hill et al., 2014), and on earlier descriptions of water droplets placed on oil-covered aluminum sheets over a cold plate, as described, for example in Vali and Stansbury (1966) and in Vali (1995). FINC has a temperature-controlled ethanol cooling bath (LAUDA Proline RP 845, Lauda-Königshofen) (Fig. 1a) in which a commercially available chip-on-board LED-array (50W COB Panel Light, Cooleeon Lighting Tech) is submerged 20 cm deep (Fig. 1b-1). A thin polytetrafluoroethylene sheet (Fig. 1b-2) acts as a diffuser mounted at a distance of 2 cm above the light source. A camera (IMX179 CMOS 8MP, ELP Free Driver) (Fig. 1b-3) mounted above the bath, images clear Piko™ PCR trays made of polypropylene (SPL0960, Thermo Fisher Scientific) (Fig. 1b-4) resting on the frame at the bath's surface. A level sensor (LLE102000, Honeywell) (Fig. 1b-5) measures the height of the ethanol in the bath and a peristaltic pump (KAS-S10-SE, Kamoer) (Fig. 1b-6) controlled by a micro-controller (Leonardo, Arduino®) (Fig. 1b-7) and powered by a stepper motor driver (TB6612, Adafruit) (Fig. 1b-7) moves ethanol between the Lauda chiller bath and the ethanol reservoir (Fig. 1b-8) (Fig. 1c). A transparent 6 mm thick polymethyl methacrylate (Plexiglas) plate (Fig. 1b-9) covers the Piko PCR trays to avoid contamination of the wells, to minimize evaporative loss of ethanol, and to limit condensation of water vapor into the ethanol (see Sect. 4.4.2). The components are mounted to a removable aluminium frame with stainless steel rods, recommended to avoid corrosion over time (Fig. 1a,b). Consumer grade hardware was used as building components where possible to reduce the overall instrument cost while ensuring measurement accuracy and reproducibility. The bulk of the building costs are constrained to the ethanol cooling bath. The per-measurement cost is dominated by laboratory consumables, such as the Piko PCR trays.

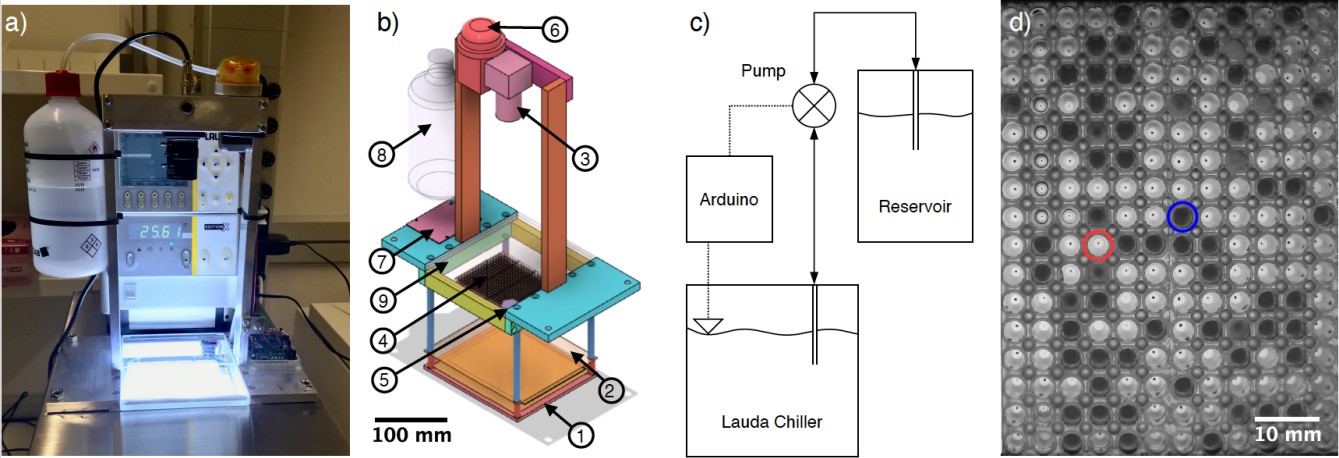

**Figure 1. (a)** Photograph of FINC. **(b)** Computer-aided design (CAD) software model of the aluminium and stainless steel movable structure placed inside the Lauda bath. The circled numbers correspond to the following parts: (1) a chip-on-board LED-array, (2) a thin polytetrafluoroethylene sheet, (3) a camera, (4) three clear polypropylene Piko™ PCR trays, (5) a level sensor, (6) a peristaltic pump, (7) an Arduino board and a stepper motor driver, (8) an ethanol reservoir, and (9) a Plexiglas plate. **(c)** Flow chart of the bath leveler setup. **(d)** Image of two of the three trays taken by the FINC camera showing the difference in light intensity between the liquid (circled in red) and frozen (circled in blue) wells used for freezing temperature detection.

### 2.1.2 PCR trays

The use of Piko PCR trays in FINC is an updated feature of existing DFTs (Table 1). Specifically, standard PCR trays contain 96 wells of 200 µL volume with dimensions of 127.76 mm by 85.34 mm, whereas the Piko PCR trays used in FINC contain 96 wells of 65 µL volume with dimensions of 80.55 mm by 26.75mm. The smaller dimensions allow for the use of up to four 96-well Piko PCR trays instead of one standard PCR tray, resulting in improved freezing temperature statistics per experiment. The trays are heated in an oven at 120 °C for at least one hour before use; this procedure improves reproducibility of background water experiments (Fig. S3). In FINC, we use three Piko PCR trays to optimize the ethanol circulation and temperature spread across the 288 sample wells (see Sect. 3.2).

### 2.1.3 Bath leveler

An automated bath leveler system was built to account for the temperature dependence of the density of ethanol in FINC's cooling bath. For example, ethanol's density at 0 °C and at −25 °C is 806.7 g L$^{-1}$ and 826.3 g L$^{-1}$, respectively. This increase in density at lower temperatures consequently translates to a decrease in volume in the cooling bath and corresponds to a 4.9 mm decrease in height, equivalent to approximately 300 mL of ethanol, between 0 and −25 °C. To achieve reproducible measurements, the ethanol level must submerge the well throughout the experiment to avoid the formation of vertical temperature gradients within the well (David et al., 2019). In FINC, a constant ethanol level is maintained by adding and removing ethanol

to the cooling bath via a peristaltic pump (Fig. 1c), thereby automating this process. The binary level sensor outputs either a submerged or emerged status signal to the micro-controller, which then turns the pump in the corresponding direction: either moving ethanol from the reservoir to the cooling bath during the experiment, or removing ethanol from the bath back to the reservoir during warm-up at the end of the experiment (Fig. 1c). The leveler does not need to be adjusted depending on the well volume used, as capillary action between the wells of the Piko PCR tray ensures that all wells are submerged in ethanol, as long as the ethanol reaches the bottom of the wells. In all, freezing measurements in FINC occur without manual intervention between measurements except for removing and placing new Piko PCR trays.

## 2.2 Cooling rate

The cooling bath temperature is controlled by a MATLAB® script, ramping down at $-1$ °C min$^{-1}$ while the script records an image every $0.2$ °C. We prefer to record images as a function of temperature rather than as a function of time to ensure that all measurements have identical increments and can be easily averaged without interpolation. The cooling rate has been previously reported to have a negligible effect on immersion freezing temperatures (Wright et al., 2013), although much faster cooling can lead to temperature assignment uncertainties (Mason et al., 2015). Thus, we chose a cooling rate matching atmospheric updraft velocities and within the capacity of the Lauda bath's cooling mechanism.

## 2.3 Freezing detection

The freezing of the solutions inside the wells of the Piko PCR trays is detected by a change in light intensity passing through the wells. Light passes through liquid water, but is scattered by ice, resulting in dark pixels in the image (Fig. 1d). Once the images are recorded over the course of one experiment, a Circular Hough transform algorithm, described in David et al. (2019), is implemented to locate the wells on the images (Fig. S1). In case of failure of the automatic well detection, we also developed a filtered algorithm output to identify the well positions, as well as a manual well alignment grid, by selecting two wells in opposing corners. After determining the well positions, the average pixel intensity is calculated for each well per image. This data analysis generates an intensity profile as a function of temperature. Then, the greatest change in intensity is attributed to the freezing temperature and can be visualized as a temporal map (Fig. 2). For example, multiple changes in light intensity for one well over the course of an experiment or neighbouring wells all freezing at the same temperature can flag an error (for example an object in front of the camera), requiring manual deletion of some images or a rerun of the measurement. A color map of the freezing temperature is also generated to visually inspect any well location bias (Fig. S2) (David et al., 2019). These data verification steps increase confidence in the measurement.

The data output of FINC is a vector containing the freezing temperature of each well. The vector is sorted by well column from top to bottom and left to right (see Fig. S1). In addition, a frozen fraction graph can be plotted by sorting the freezing temperature vector and plotting it versus a linearly spaced vector with values ascending from 0 to 1 in 288 steps (example in Fig. S3). Based on the recommendation by Polen et al. (2018), the data is not trimmed, and all 288 freezing temperature data points are plotted. These data manipulations retain all information available from the experiment: freezing temperature and well location in one vector.

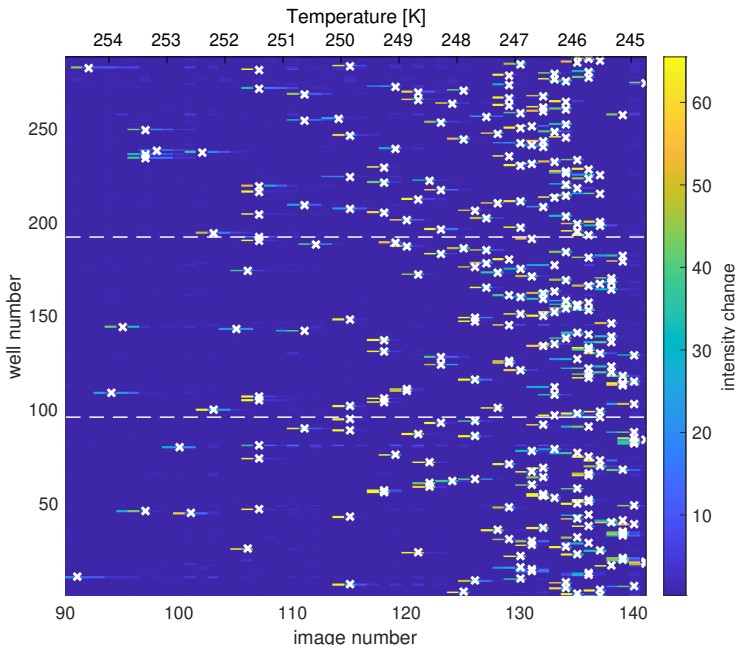

**Figure 2.** Map of the change in light intensity for each of the 288 wells (y-axis) as a function of the image number (x-axis), recorded every 0.2 °C and corresponding to temperature decreasing and experimental time progressing. (The first 90 images corresponding to an interval of 18 degrees are excluded here for simplicity). The color scale represents the first derivative of the light intensity, with bright colors indicating a sudden drop in light intensity associated with freezing of the well. The algorithm assigns the freezing temperature of the well to the greatest change in light intensity and is marked by a white "x". The dashed horizontal lines indicate the tray boundaries between the three Piko PCR trays. The freezing experiment depicts the results of a background water measurement.

## 2.4 Sample preparation and data analysis

Sample solutions were prepared within a laminar flow hood (Labculture Class II BioSafety Cabinet, ESCO) to prevent contam-
ination from lab air. All glassware were pre-rinsed three times with deionized water and with acetone, and subsequently dried
in an oven at 120 °C for at least one hour. Sample solutions were then prepared using molecular biology-free reagent water
(W4502, Sigma Aldrich, hereafter referred to as background water). The solutions were then pipetted from sterile plastic reser-
voirs (10141-922, VWR) with an electronic multi-pipettor (4671040BT, Thermo Fisher Scientific, USA) into pre-cleaned Piko
PCR trays (see Sect. 2.1.2). We recommend pipetting the solution volume in one dispensing volume to minimize the formation
of bubbles (see 4.4.4). In addition, Barry et al. (2021) recently suggested working with cleanroom vinyl or polyethylene gloves
and aluminium foil coated surfaces.

When transporting the Piko PCR trays from the laminar flow hood to FINC, we placed cover films (Z369667, Sigma Aldrich)
over the trays to avoid contamination from ambient air. We removed the cover film before placing the trays inside the ethanol

bath, in order to avoid problems with ethanol penetrating between the cover film and the Piko PCR tray by capillary effects.
Furthermore, the plexiglass hinged-cover over the trays prevents contamination from the air depositing into the wells.

Note that our sample preparation procedure does not include dilution series; we made the solutions with the required concentration from weighed solid. We have previously proposed that lignin may be aggregating in solution, leading to concentration-dependant ice nucleation behaviour of INMs (Bogler and Borduas-Dedekind, 2020). Based on this hypothesis, we also do not conduct any data merging procedure for different concentrations of INMs or drop volumes (for example in Figs. S6  S7). Furthermore, we do not subtract background water values, and prefer to show raw data in box plot formats (as in Fig. 7 and see also Brennan et al. (2020)).

## 3  FINC's uncertainties

### 3.1  Temperature uncertainty

A temperature calibration is necessary to correct for the difference between the recorded temperature of the Lauda bath and the temperature within each well. This calibration was done using a multi-channel thermocouple data logger (HH-4208SD, Thermosense, UK), where each K-type thermocouple was placed inside a well filled with ethanol, with nine thermocouples evenly spaced across the three trays. With the probes inside the wells, the bath temperature was ramped down to $-30\,^\circ$C at a rate of $-1\,^\circ$C min$^{-1}$. The temperature of each thermocouple was recorded every 10 seconds ($T_{\text{well}}$) and was plotted against the bath temperature recorded by the Lauda system ($T_{\text{bath}}$), as shown for one calibration experiment in Figure S4. We then obtained the following calibration equation (Eq. 1), where the slope and intercept are averaged values across three independent calibration experiments:

$$T_{\text{well,avg}} = 0.95 * T_{\text{bath}} + 0.75 \tag{1}$$

Multiple temperature calibrations conducted several months apart and by multiple users led to identical slopes, confirming reproducible temperature gradients and constant ethanol circulation inside the bath. The mean of the standard deviations of the nine evenly-spaced thermocouples across three Piko PCR trays was $0.5\,^\circ$C for temperatures down to $-25\,^\circ$C (Table S1). We therefore report the temperature uncertainty for each well to be $\pm\,0.5\,^\circ$C.

### 3.2  Temperature spread across wells

Furthermore, we tested different numbers of Piko PCR trays as well as different bath pump speeds, to reduce the temperature bias across the three trays within an experiment (Sect. S3, Table S1). Tests with four trays led to the identification of a vortex in the upper left corner of the ethanol bath, yielding higher temperature biases across the trays (Table S1). Overall, we determined that using three trays, placed to the right to avoid the vortex, as well as pump speed setting 8, resulted in the smallest temperature biases of $\pm0.46\,^\circ$C between $-10$ and $-15\,^\circ$C and of $\pm0.55\,^\circ$C between $-20$ and $-25\,^\circ$C (Table S1, last row entry). Since the average of these values is precisely $0.5\,^\circ$C, we consider that this value is equal to the temperature uncertainty described in Section 3.1.

## 3.3 FINC's limit of detection

It is necessary to accurately characterize the background of the instrument to determine the lowest trustworthy freezing temperature of a sample. FINC's limit of detection (LOD) for the freezing control experiments (Sect. 4) was calculated as the mean of ten replicates of background water experiments using 5 μL droplets (Fig. S5). We calculated the mean temperature and one standard deviation (a spread of 1 $\sigma$) for each of the 288 values, resulting in an LOD $T_{50}$ of $-25.4 \pm 0.2$ °C. We note that a value of $\pm 3 \sigma$ can also be used and would lead to a similar background $T_{50}$ of $-25.4 \pm 0.4$ °C (Fig. S5). Furthermore, we show the LOD as a boxplot of the mean frozen fraction (Fig. S5). The LOD depends on the experiment type, but as long as the appropriate background characterizations are measured, e.g. artificial salt water, background water, water through a laboratory setup, etc., the instrument can be used to measure freezing temperatures of 288 wells at a time (see Sect. 4.5 for further discussion). Finally, we add that no background corrections are made in our data analysis.

## 4 Freezing control experiments

To test the capabilities of FINC and to characterize its sources of uncertainties, we conducted several freezing control experiments. We considered (1) the non-homogeneous freezing of the background water (Sect. 4.1 and Sect. 4.2), (2) the roles of tray material and of droplet shape (Sect. 4.3), (3) eight different sources of contamination (Sect. 4.4), (4) the choice of well volume (Sect. 4.5), and (5) the freezing-point depression of a dissolved organic matter solution with different salt concentrations (Sect. 4.6).

### 4.1 Non-homogeneous freezing in FINC

The use of Piko PCR trays allows for a range of sample volumes between $5 - 60$ μL to be measured in FINC. Theoretically, freezing rates of water droplets are dependent on the volume of the droplet; smaller droplets freeze at lower temperatures (O and Wood, 2016; Koop and Murray, 2016). Classical nucleation theory approximates interfacial tension between ice and water, activation energy of the phase transfer, and size of clusters and embryos (Ickes et al., 2015). In the atmosphere, 50% of a droplet population of 5 μL-volume is predicted to freeze spontaneously ($< 1$ s) and thus homogeneously at $-31.81$ °C, whereas 60 μL-volume is predicted to freeze at $-31.41$ °C (equations from Wang (2013)). However, these temperatures were never reached during FINC experiments with background water. Over the range of possible sample volumes in FINC (5–60 μL), the mean $T_{50}$ value of background water was $-24.5 \pm 0.8$ °C. DFTs tabulated in Table 1 which use drops in the microliter range also show this non-homogeneous freezing behaviour. As argued in the following sections, the difference in temperatures between atmospheric homogeneous freezing and background water freezing in FINC is likely due to a combination of tray material (Sect. 4.3.1), of non-spherical drop shapes within the wells (Sect. 4.3.2), and of different sources of contamination (Sect.4.4).

## 4.2 Volume dependence on non-homogeneous freezing

We further attempted to quantify these uncertainties by comparing the $T_{50}$ values of background water over a range of background water volumes. The mean $T_{50}$ background water values in FINC were $-25.4 \pm 0.1$ °C, $-24.7 \pm 0.1$ °C, $-23.4 \pm 0.6$ °C, $-24.5$ °C, $-24.4 \pm 0.2$ °C, $-25.2$ °C, and $-25.3 \pm 0.1$ °C for 5 µL, 10 µL, 20 µL, 30 µL, 40 µL, 50 µL, and 60 µL, respectively, across one to five replicates (Table S2; Fig. S6). Furthermore, if we collect all the freezing temperatures from different volumes described in Sect. 4.1 and convert the data into INPs per volume, we observed a large spread in the freezing behaviour of the water (Fig. S7). Through both these data analyses, we observe that experiments with 20 µL of background water freezes warmer and with a larger spread than all other volumes. In Sections 4.3 and 4.4, we investigate and discuss the potential sources of contamination involved in creating a volume-dependent non-homogeneous freezing temperature as well as attempt to reason the unpredictable behavior of the 20 µL freezing experiments.

## 4.3 Potential uncontrollable factors affecting non-homogeneous freezing

### 4.3.1 Effect of tray material

As described in detail in Li et al. (2012) and in Polen et al. (2018), the material interacting with the supercooled water droplets impacts their freezing. Indeed, the Piko PCR trays used in FINC are made of polypropylene and are therefore a hydrophobic surface. Interestingly, hydrophobic surfaces have previously been observed to freeze at a warmer temperature than a hydrophilic surfaces (Li et al., 2012). It is likely that the material of the tray is contributing to warmer temperatures than expected for homogeneous freezing, but it is difficult to quantify the extent or percentage of this contribution to the overall non-homogeneous freezing behavior. It remains that imperfections on the surface of each well could also induce non-homogeneous freezing behavior (Diao et al., 2011), and that the use of a larger number of wells could help provide reliable statistics.

### 4.3.2 Effect of drop shape

Due to the narrow width of the Piko PCR tray wells, the water in FINC's wells is subjected to capillary forces. Indeed, a concave meniscus is evident when examining the solution in the Piko PCR trays, thereby exerting negative pressure on the solution. It has been previously shown that the homogeneous nucleation rate of water can be significantly increased when water is subjected to negative pressure (Marcolli, 2017, 2020). However, the negative pressure associated with the radius of the meniscus in the Piko wells is on the order of 1-3 mm (see Table S3 and Fig. S10) and are likely negligible to participate in non-homogeneous freezing in FINC (Marcolli, 2017, 2020). Nevertheless, classical nucleation theory assumes spherical supercooled water droplets, whereas DFTs using a cold stage or PCR trays are rather investigating the freezing behaviour of half-spheres or cone-shaped supercooled droplets.

### 4.4 Potential controllable sources of contamination

#### 4.4.1 Effect of background water contamination

The background water is a contentious issue in the field of atmospheric ice nucleation. An excellent overview of the challenges of "cleaning up our water" is described in Polen et al. (2018). Upon their recommendation, we experimented with different types of purified water and found that the molecular biology-free reagent water (i.e. background water) gave the most reproducible measurements, consistent with David et al. (2019). Filtration of the background water through a 0.02 μm filter led to no difference in freezing behavior for the background water, except for our lab's Milli-Q water which is known to contain higher levels of organic carbon (Fig. S8). We therefore validated our choice of background water used without further purification.

#### 4.4.2 Effect of condensing water vapor

We considered the possibility of water vapor condensing into the wells from the air between the plexiglass and the trays during a measurement (Fig. S9). We calculated a maximum amount of condensable water vapor, or in other words, a worst case scenario, where if the volume of water vapour corresponding to 90% RH at 0 °C between the plexiglass and the trays were to condense into liquid water inside a well. We arrived at a value of 2 nL (Sect. S7). This volume is small and unlikely to be able to condense into one well, freeze on a colder wall and trigger nucleation within the well. We therefore conclude that this condensation process has a negligible effect on the well volume and thus on freezing temperatures.

#### 4.4.3 Effect of the surface area of the tray

We considered whether a difference in surface area to volume ratio could explain the different $T_{50}$ values observed for different volumes depicted in Figure S6 (see Sect. S8 for the calculations of surface areas of the wells). In particular, we hypothesized that the warmer freezing of the 20 μL could be due to the cone-like shape of the well (see shape diagram in Fig. S10). However, the surface area to volume ratios of the different volumes shown in Figure S11 could not explain the variability or higher freezing temperatures observed, particularly with 20 μL. Note that the water vapor condensation is inconsequential to changing the surface area to volume ratio (Sect. 4.4.2). Our running hypothesis to the peculiar higher freezing of the 20 μL experiments is the presence of microscopic bubbles generated at the intersect of the two truncated-cones when using this volume (Sect. 4.4.4).

#### 4.4.4 Effect of air bubbles in the wells

Air bubbles in the solutions within a well can be generated during the pipetting process, particularly if the solution mixture has surfactant-like properties (see example image in Fig. S12). Bubbles are defined here as visible pockets of air in the well. This problem is likely the result of a combination of trapped air in the narrow wells during dispensing. Wells containing bubbles in FINC (Fig. S12) froze at warmer temperatures (Fig. S13). When the bubbles collapsed within the well, they may have created a spike in negative pressure, inducing freezing at warmer temperatures than in the absence of bubbles (Marcolli,

2017). Therefore, bubbles should be avoided by careful introduction of the solution into the well. Since bubbles associated with pipetting can be seen in the images, we can confirm the absence of bubbles in all images used for the data plotted in Figure S6. As no visible bubbles were present, then this potential effect is unlikely to explain the observed non-homogeneous freezing behavior for different well volumes. Nevertheless, microscopic bubbles invisible to the human eye and to the camera could be affecting the non-homogeneous freezing, including for the 20 μL volumes, but to an unknown degree.

### 4.4.5 Effect of contamination from the tray

A series of experiments were conducted to test the leaching of potential ice-active material from the Piko PCR trays into the background water. To test this hypothesis, we placed 60 μL of background water into one tray and then pipetted 50 μL of the water out of each well and into another tray. If leaching was indeed a problem, we would have observed higher freezing temperatures in the tray containing the transferred solution. However, we found no effect to freezing from leaching over duplicate sets of experiments (Fig. S14). We conclude that material leaching from the Piko PCR trays is not a problem.

### 4.4.6 Effect of lab air contamination

As discussed by Whale et al. (2015) and by Stopelli et al. (2014), the concept of an open droplet system can be prone to further contamination from the surrounding air. We have built upon the authors' recommendations by using a laminar flow hood, by placing a cover film on the top of the Piko PCR tray between the flowhood and FINC, and by having a plexiglass cover on FINC to minimize deposition of airborne contaminants. Furthermore, we calculated the surface area of the solution exposed to air, but observed no trend related to the non-homogeneous freezing behavior for different well volumes (see Table S3).

### 4.5 Volume of solution per well discussion and recommendation

Based on Sections 4.1, 4.2, 4.3 and 4.4, it remains difficult to quantify the contribution of each possible source of contamination to non-homogeneous freezing. This difficulty appears to be a combination of uncontrollable (Sect. 4.3) and controllable factors (Sect. 4.4), and has been previously addressed in Polen et al. (2018). Nevertheless, the pre-treatment of the trays (see Fig. S3), the use of a laminar flow hood, and the use of molecular biology-free reagent water allowed for reproducible background water measurements. In all, the recommended well volume while working with FINC depends on the research question. One should use a larger volume to study active, but less abundant, INPs and INMs, whereas one should use a smaller volume to study less active, but more abundant, INPs and INMs. If the samples have high concentrations of salts leading to freezing point depression, for instance, then a larger volume might be necessary to remain in the operating range of the Lauda bath.

### 4.6 Freezing-point depression

Inorganic salts have a freezing-point depression effect on water's freezing temperature (e.g., Koop et al., 2000b). The effect is described in general by the relationship in Eq. 2, commonly known as Blagden's Law:

$$\Delta T_f = K_f * b_b \tag{2}$$

where $\Delta T_f$ is the freezing-point depression, $K_f$ is the cryoscopic constant of the solvent ($K_{f,\text{water}} = 1.86$ K kg mol$^{-1}$), and $b_b$ is the molality of the solute (Atkins and de Paula, 2011). To further characterize FINC, we tested the ice nucleation activity of solutions of dissolved organic matter (DOM) and of sodium chloride (NaCl; 31434, Sigma Aldrich). The DOM solutions

were at concentrations of 20 mg carbon per liter (mg C L$^{-1}$) and were samples obtained from Jericho Ditch, part of the Great Dismal Swamp in Suffolk, Virginia, USA (sample collection reported in Borduas-Dedekind et al., 2019). Jericho Ditch DOM is ice active, with a $T_{50}$ value of $-10.6 \pm 0.0$ °C (Fig. 3), and is therefore a suitable sample for freezing-point depression tests. We determined a linear freezing-point depression effect with increasing concentrations of salt, as expected by the theoretical values calculated according to Blagden's Law (Fig. 3; Eq. 2). Our saline DOM solutions measured with FINC match well with

Blagden's Law. However a small deviation was observed with a 3 M NaCl solution, likely due to activity coefficients deviating from unity. This experiment further validates the instruments capabilities as a droplet freezing technique (Fig. 3).

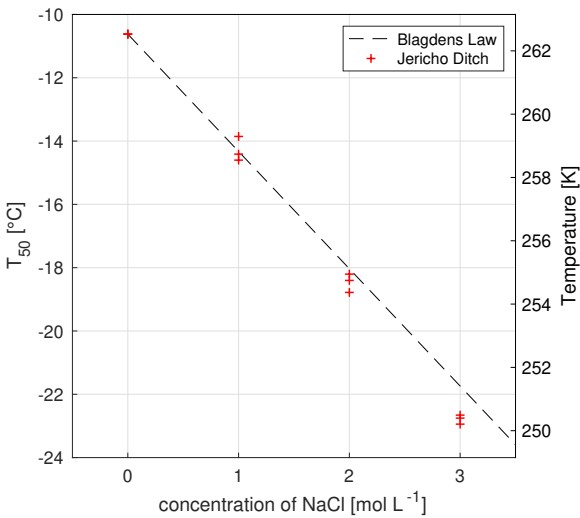

**Figure 3.** The freezing-point depression effect of solutions of 20 mg C L$^{-1}$ solution of Jericho Ditch dissolved organic matter with increasing concentrations of sodium chloride (NaCl). The x-axis represents the concentration (mol L$^{-1}$) of NaCl, and the y-axis represents the $T_{50}$ values, where 50% of the wells were frozen. Triplicate experiments were done for each of the concentrations of 0, 1, 2, and 3 M NaCl, marked with red crosses. Note that the $T_{50}$ values for the experiments at 0 mol L$^{-1}$ are identical and thus overlap. Experimental data was compared to Blagden's Law (grey dashed line), which describes the freezing-point depression phenomenon. All temperature measurements have an uncertainty of $\pm 0.5$ °C.

## 5    Drop freezing instrument intercomparison and validation of FINC

To validate FINC against other similar and peer-reviewed DFTs, we conducted an intercomparison study with ETH's DRINCZ (David et al., 2019) and University of Basel's LINDA (Stopelli et al., 2014) (Sect. 5.1). Here, we report the comparison

measurements using NX-illite (Sect. 5.2) and an ambient aerosol sample (Sect. 5.3).

## 5.1 Experimental details of DRINCZ and of LINDA

DRINCZ was operated using a freshly shaken sample poured into a sterile reservoir. Then, 96 droplets of 50 μL were transferred into a 96-well PCR polypropylene tray (732-2386, VWR, USA) using an 8-channel multi-pipette. The tray was sealed with a transparent foil and immediately analyzed with DRINCZ as described in David et al. (2019). LINDA was operated using a total of 5.2 mL per experiment and each sample was shaken by hand and immediately pipetted under a laminar flow hood (AURA Mini, EuroClone) into 52 microtubes (0.5 mL Eppendorf Safe-Lock Tubes) using a repeater pipette (Stepper TM 411, Socorex) and bioproof syringes (Ecostep TM, sterilized, single wrapped, bioproof, range: 50 – 500 μL). Each droplet contained 100 μL of the sample solution, and was measured with LINDA, according to Stopelli et al. (2014).

All samples meant for intercomparison were prepared at ETH Zurich on July 10, 2019. On the following day, measurements at all three instruments were conducted. As LINDA was located in Basel, one batch of aliquots was transported in a cooler by train in the morning from the preparation location (in Zurich) to the measurement location (in Basel). For background measurements, purchased, bottled molecular biology-free reagent water (Sigma-Aldrich, W4502-1L) was used (background measurements reported in Figure S15), and no background corrections were made.

## 5.2 NX-illite intercomparison

NX-illite has been repeatedly used as a standard to compare ice nucleation instruments (e.g., Hiranuma et al., 2015a). Commercial NX-illite solutions of 0.01, 0.005, and 0.001 %wt (or 0.1, 0.05, and 0.01 g of NX-illite per 1 L of water, respectively) were measured by FINC, DRINCZ, and LINDA during an intercomparison experiment day on July 11, 2019 (Fig. 4a). The three instruments measured the same suspension; one suspension of each concentration was split into several sterile Falcon tubes (14-432-22, Thermo Fisher Scientific, USA). The Falcon tubes were shaken immediately prior to pipetting, and the filled PCR trays were not left to sit more than a few minutes prior to the freezing experiment. We calculated, according to Eq. 3 (Vali, 1971, 2019), the ice active surface-site density ($n_{s,\text{BET}}$), where BET stands for the Brunauer–Emmett–Teller technique, a commonly used technique to measure particle surface areas (Brunauer et al., 1938):

$$n_{s,\text{BET}}[\text{m}^{-2}] = -\frac{\ln[1 - \text{FF}(T)]}{SA_{\text{BET}} * C_{\text{illite}} * V_{\text{well}}} \tag{3}$$

where $\text{FF}(T)$ is the frozen fraction at each freezing temperature, $SA_{\text{BET}}$ is the BET-determined surface area of the NX-illite particles ($124.4 \text{ m}^2\text{g}^{-1}$, (Hiranuma et al., 2015a)), $C_{\text{illite}}$ is the mass concentration of NX-illite, and $V_{\text{well}}$ is the volume in each well ($V_{\text{well}}$ = 30 μL for FINC, 50 μL for DRINCZ, and 100 μL for LINDA). The uncertainties in $n_{s,\text{BET}}$ include a 1% error in $SA_{BET}$ (Broadley et al., 2012), 1% error in $C_{illite}$, 1% error in $V_{\text{well, DRINCZ}}$, 0.5% error in $V_{\text{well,LINDA}}$, 8% error in $V_{\text{well,FINC}}$ (well volume errors are based on error in pipettes), and an error of $\pm$ 1 in the number of frozen wells. Uncertainty calculations are detailed in Section S11, and plotted in Figure 4a as vertical error bars on the $T_{50}$s for one sample from each instrument.

We additionally compare these measurements with NX-illite solutions measured on DRINCZ in 2018 (David et al., 2019), as well as with the parameterization from Hiranuma et al. (2015a) (Fig. 4a).

The FINC measurement spread at −15 °C is from $68 - 209 \text{ m}^{-2}$, and grows to $2600 - 9200 \text{ m}^{-2}$ at −20 °C. The Hiranuma et al. (2015a) parameterization fits well within the FINC measurements above −17 °C, and deviates up to a factor of 4.8 at

−21 °C. Between FINC, DRINCZ, and LINDA, the spread was a factor 6 at −15 °C and a factor 7.5 at −20 °C. The DRINCZ measurements from David et al. (2019) are up to one order of magnitude higher than those reported here (Fig. 4a). Indeed, the spread of $n_{s,\mathrm{BET}}$ is a common outcome of NX-illite suspensions (e.g. David et al., 2019; Harrison et al., 2018; Beall et al., 2017). We hypothesize that this spread is due to the heterogeneity of the suspensions; the NX-illite particles can settle and sediment to the bottom of the wells, reducing the available surface area to nucleate ice. Furthermore, sedimentation increases with concentration, consistent with the observation of lower $n_{s,\mathrm{BET}}$ values at higher concentrations (Fig. 4a). To further test this hypothesis, we retested the 0.1 g L$^{-1}$ solution on FINC eight months later, and found that the $n_{s,\mathrm{BET}}$ values had decreased by approximately a factor of 5 at −20 °C (Fig. 4a, sample FINC Retest in blue). This change in $n_{s,\mathrm{BET}}$ values suggests that NX-illite suspensions are not stable over time, consistent with mineral dust experiments demonstrating ion exchange abilities in solution over time (Kumar et al., 2019). Nevertheless, the results of this intercomparison with NX-illite supports the validation of FINC as a suitable instrument for quantifying ice nucleating activity in the immersion freezing mode, yet strengthens our proposal for a more solution-stable standard.

## 5.3 Ambient aerosol intercomparison

Next, we measured the INP concentration of an ambient aerosol sample during the intercomparison study with DRINCZ and LINDA. Ambient aerosols were collected with a Coriolis μ air sampler (Bertin Technologies, France), an instrument designed for outdoor monitoring of bio-aerosols such as pollen and spores (Gómez-Domenech et al., 2010; Carvalho et al., 2008). The sample was collected for $t_{\mathrm{aq}} = 20$ minutes at a flow rate of $Q = 300$ L min$^{-1}$ on July 10, 2019 on the terrace of the Institute for Atmospheric and Climate Science at ETH Zurich in Switzerland. The initial sample volume $v = 15$ mL was further diluted with a dilution factor of $DF = 3.7$ to obtain enough volume to split into sterile Falcon tubes and brought to FINC, DRINCZ, and LINDA for measurement the following day. The INP concentration per L of air was calculated by (Eq. 4):

$$\mathrm{INP}[\mathrm{L}_{\mathrm{air}}^{-1}] = -\frac{\ln[1 - \mathrm{FF}(T)]}{V_{\mathrm{well}}} * \frac{DF * v}{t_{\mathrm{aq}} * Q} \tag{4}$$

The uncertainties associated with the INP concentrations are plotted in Figure 4b as vertical error bars on the $T_{50}$s for one sample from each instrument, and the calculations are described in Section S11. The uncertainties include 1% error in $V_{\mathrm{well, DRINCZ}}$, 0.5% error in $V_{\mathrm{well,LINDA}}$, 8% error in $V_{\mathrm{well,FINC}}$ (well volume errors are based on error in pipettes), as well as an error of $\pm 1$ in the number of frozen wells, an error in the flow rate of the Coriolis impinger of $\pm 10$ L min$^{-1}$, and an error of $\pm 0.5$ mL in final Coriolis sample volume.

The INP concentration in the aerosol sample as a function of temperature agreed well between the instruments, with the advantage of FINC being able to achieve lower temperature measurements than DRINCZ and LINDA due to the smaller well volume (Fig. 4b, and see Sect. 4.5). The measurements also fit within the range of INP concentrations per L of air from cloud water and precipitation samples, compiled by Petters and Wright (2015). Furthermore, since LINDA was part of the intercomparison study of ice nucleation instruments reported by Hiranuma et al. (2019) which used cellulose as a standard, FINC and DRINCZ are by extension expected to have representative freezing temperature ranges. These results further validate the capabilities of FINC as a DFT.

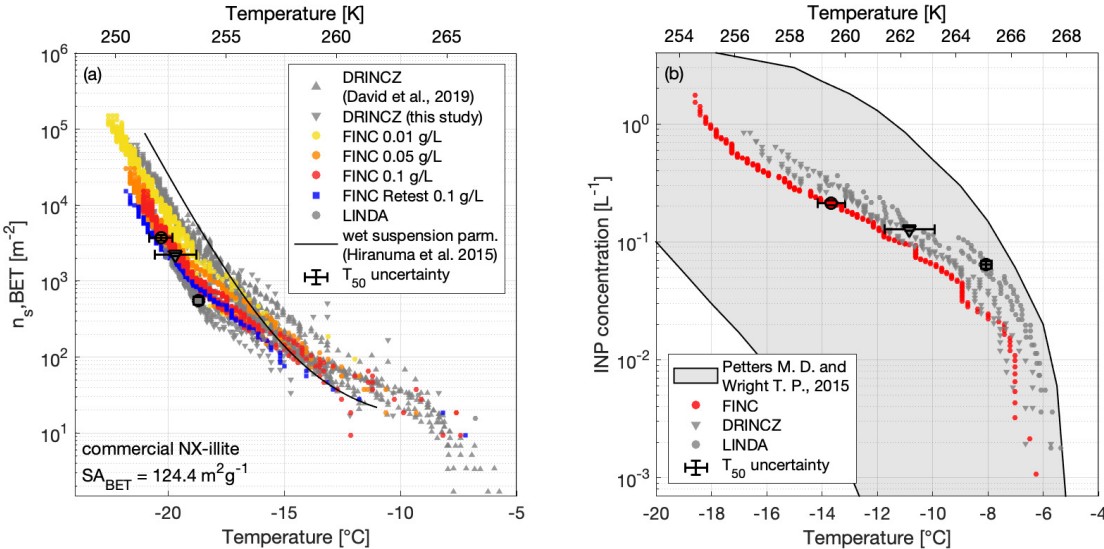

**Figure 4. (a)** The active surface-site density $n_{s,\text{BET}}$ of NX-illite suspensions measured on FINC (yellow, orange, red dots), DRINCZ (grey downward triangles), and LINDA (grey dots) during the intercomparison. The wet suspension parameterization by Hiranuma et al. (2015a) is plotted for reference. Concentrations used were 0.1, 0.05, and 0.01 g $L^{-1}$ of NX-illite. The DRINCZ illite measurements as reported in David et al. (2019) are also shown for comparison (grey upwards triangles). $n_{s,\text{BET}}$ was calculated using a BET surface area of 124.4 $m^2 g^{-1}$ (Hiranuma et al., 2015a). Error bars in the x- and y-direction are shown on the $T_{50}$ value for one sample each of DRINCZ, LINDA, and FINC (see Supplemental Section 11 for details). Note that the y-direction uncertainties do not extend beyond the marker. **(b)** INP concentrations ($L^{-1}$) of an ambient aerosol sample solution from Zurich, Switzerland, measured on FINC (red dots), DRINCZ (grey triangles), and LINDA (grey dots). Ambient aerosols were collected with a Coriolis $\mu$ impinger for 20 minutes at 300 L $\min^{-1}$. The sample was split for measurements on each of the immersion freezing instruments. The grey shaded area denotes the range of INP concentrations from precipitation and cloud water samples compiled by Petters and Wright (2015). Error bars in the x- and y-direction are shown on the $T_{50}$ value for one sample each of DRINCZ, LINDA, and FINC (see Supplemental Section 11 for details). Note that the y-direction uncertainties do not extend beyond the marker.

## 6  Lignin as an ice nucleation standard

We describe here the use of commercial and water-soluble lignin as a standard for intercomparing droplet freezing techniques (DFTs). Its applicability is demonstrated through the intercomparison measurements of lignin with three different DFTs (Sect. 6.2), the comparison of different batches of a commercial lignin products (Sect. 6.3), the stability of lignin solutions over time (Sect. 6.4), and the recalcitrance of lignin towards chemical and atmospheric processing (Sect. 6.5).

**Figure 5.** Example of the chemical structure of a lignin monomer, known as a lignol. The commercial kraft lignin material used in this work has a thiol group in the benzylic position instead of an alcohol group (highlighted in bold).

## 6.1 Chemical composition of lignin

Lignin is a high molecular weight natural polymer accounting for approximately 30% of the organic carbon present in the biosphere (Boerjan et al., 2003). It is composed of three different types of hydroxycinnamyl alcohol monomers referred to as monolignols, specifically *para*-coumaryl alcohol, coniferyl alcohol and sinapyl alcohol (Faraji et al., 2018) (Fig. 5). The relative amount of monolignols within the lignin polymer vary depending on taxa, cell type and cell layers, as well as on the development stage of the tree, the climate and the habitat (Boerjan et al., 2003), leading to a large number of possible permutations within the natural lignin polymer. The polymerisation of monolignols occurs through a stepwise chemically-controlled process linking ether bonds ($\beta$-O-4, $\alpha$-O-4) and carbon-carbon bonds (Ralph et al., 2019). Lignin indeed constitutes a mixture of monolignols but specifically represents repeating units of known chemical functional groups such as oxygen-substituted arenes, conjugated double bonds and alcohols. Lignin's Van-Krevelen diagrams determined using high resolution mass spectrometry, typically has H:C ratios between 0.6-1.6 and O:C ratios between 0.1 and 0.7 (Devarajan et al., 2020). These ratios are distinct from other classes of organic matter, namely carbohydrates, lipids, peptides and tannins (Kim et al., 2003; Hockaday et al., 2009; Ohno et al., 2010; Sleighter et al., 2010). Interestignly, lignin has recently been modeled to aggregate with itself (Devarajan et al., 2020). Furthermore, the commercial lignin used in this work is a by-product of the pulp and paper industry and therefore has a reproducible chemical composition, arguably important for a standard, and a molecular weight average of 10 000 g mol$^{-1}$. Section 6.3 confirms this reproducibility specifically for ice nucleation. We further acknowledge that this commercial lignin contains a thiol group, which differs from atmospheric relevant monolignols, yet is used here as a soluble and cheap standard for the intercomparison of DFTs.

## 6.2 Instrument intercomparison with lignin and IN parameterization

The measurements on FINC, DRINCZ, and LINDA were conducted on the same day (July 11, 2019) with identical lignin solutions. Specifically, one solution of 20 mg C L$^{-1}$ lignin (of Batch 1, see Sect. 6.3) diluted in background water was divided into sterile Falcon tubes and brought to each instrument.

We calculated the ice-active mass site density, $n_m$ values, for each measurement according to Eq. 5 and analogous to Eq. 3, also following from Vali (1971, 2019):

$$n_m[\text{mg}^{-1}] = -\frac{\ln[1 - \text{FF}(T)]}{\text{TOC} * V_{\text{well}}} \tag{5}$$

where TOC is the total organic carbon of the lignin solution, and $V_{well}$ is the volume inside each well ($V_{well}$ is 30 μL for FINC during the intercomparison study, 5 μL for FINC during batch experiments, 50 μL for DRINCZ, and 100 μL for LINDA). TOC values were determined using the mass of lignin weighed and the vendor's description of carbon content (50.13%) and were further validated using a Shimadzu TOC analyzer with the prior pre-treatment addition of nitric acid and sulfuric acid. Uncertainties in $n_m$ include a 1% error in TOC, 1% error in $V_{\text{well, DRINCZ}}$, 0.5% error in $V_{\text{well,LINDA}}$, 8% error in $V_{\text{well,FINC}}$ (well volume errors are based on error in pipettes), and finally an error of $\pm 1$ in the number of frozen wells. The uncertainty calculations are presented in Section S11 and are displayed in Figure 6 as error bars on the $T_{50}$ of one sample from each instrument.

The freezing temperatures of 20 mg C L$^{-1}$ compared well between FINC, DRINCZ, and LINDA, with overlapping $n_m$ traces, falling within a factor of 3 (Fig. 6). Specifically, at $-14.8$ °C, lignin's $n_m$ values from the intercomparison range from 77 mg$^{-1}$ to 223 mg$^{-1}$, and at $-19.5$ °C the range is 1760 to 4560 mg$^{-1}$. This spread is an improvement relative to NX-illite, which has $n_s$ values that span an order of magnitude (Fig. 4a). Moreover, the data from the intercomparison was fitted to obtain a parameterization for lignin's ice nucleating ability in immersion freezing, shown as the solid black line in Figure 6. In addition, $T_{50}$ and $T_5$ values were $-17.5 \pm 0.9$ and $-12.4 \pm 1.3$ °C, respectively. Unlike Kunert et al. (2018) who observed reproducible initial freezing temperatures with Snowmax, we observe lignin's freezing temperatures being reproducible starting at $T_5$ values (where 5% of wells are frozen). Note that the FINC batch measurements, also shown in Figure 6, were not included in the empirical fit to avoid over-weighing the FINC data. The fit is represented in Eq. 6, where $T$ is temperature in °C:

$$n_m = \exp(-0.558 * T - 3.12) \tag{6}$$

Note that this parameterization is specifically valid for lignin solutions of 20 mg C L$^{-1}$. In fact, lignin is a macromolecule suggested to adopt different solution aggregation properties depending on its concentration (Bogler and Borduas-Dedekind, 2020), and thus our paramaterization is exclusively for concentrations of 20 mg C L$^{-1}$, equivalent to 40 mg of lignin per L. For reference to other INMs, the $n_m$ parameterization by Wilson et al. (2015) of sea-surface microlayer organic matter is included (Fig. 6). While the slope is similar for both parameterizations (Fig. 6), the Wilson paramaterization is interestingly two orders of magnitude higher. Lignin is not the most active INM in the sea surface microlayer, in sea spray, or in bioaerosols (Steinke et al., 2020), and so we were therefore surprised to notice the similarity in slope. However, organic matter $n_m$ data from several other studies are inconsistent with this slope, highlighting the difficulties in predicting the ice nucleating ability of organic matter (Borduas-Dedekind et al., 2019; Pummer et al., 2015; O'Sullivan et al., 2014; McCluskey et al., 2017).

### 6.3 Lignin batch comparison

A desirable quality of a suitable ice nucleation standard is its ability to be reproducible across commercial product batches. The lignin used here is alkali low-sulfonate kraft lignin (CAS 8068-05-1, Sigma Aldrich, product code 471003). We tested the ice

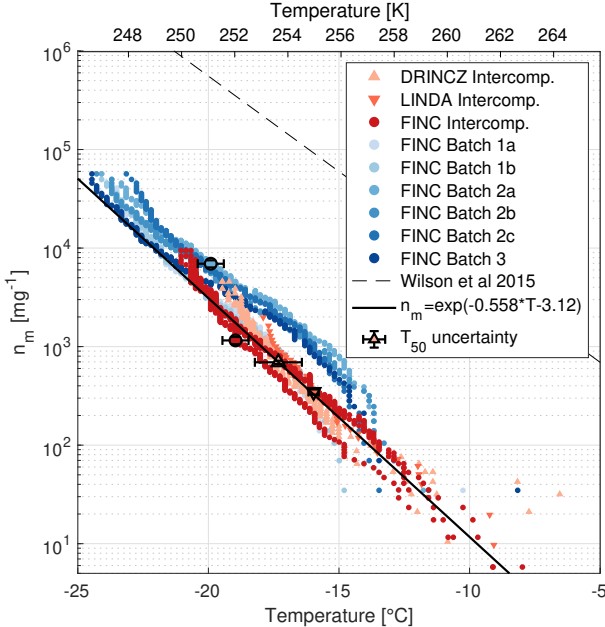

**Figure 6.** Ice active mass site density ($n_m$) of lignin measured by FINC, DRINCZ, and LINDA. Lignin was measured during the inter-comparison on FINC (30 μL well volume; 3 replicates; dark red dots), DRINCZ (50 μL well volume; 4 replicates; orange triangles), and LINDA (100 μL well volume; 1 replicate; red triangles). Additionally, different product batches of lignin were measured by FINC (5 μL well volume; 6 experiments, blue dots). Note that the markers of Batch 1a and 1b are partially hidden behind the orange intercomparison markers. All lignin solutions were of concentration 20 mg C L$^{-1}$. The black line is the empirical fit of the intercomparison data only ($n_m = exp(-0.558 * T - 3.12)$). The parameterization of the sea-surface microlayer organic matter from Wilson et al. (2015) is also shown (dashed lined). Error bars in the x- and y-direction are shown on the $T_{50}$ value for one sample each of DRINCZ, LINDA, FINC batch experiments, and FINC intercomparison experiments (see Supplemental Section 11 for details). Note that the y-direction uncertainties do not extend beyond the marker.

nucleating ability of three different batches of this product: batch numbers 04414PE, MKCL2371, and MKCK3344 (hereafter referred to as Batch 1, 2, and 3, respectively). Specifications of their production dates and their carbon and sulfur contents are provided in Table S4. The $n_m$ values, determined according to Eq. 5, of Batch 2 and 3 overlapped with each other (Fig. 6). The

values of Batch 1 overlapped with the other batches below −19 °C, but above −19 °C had fewer ice active mass site densities (Fig. 6). Still, all batches were within one order of magnitude of each other and of the empirical fit from the intercomparison data (Fig. 6). These results indicate that lignin's ice nucleating activity is reproducible across different production batches.

### 6.4 Lignin solution stability

In addition to reproducibility across product batches, another quality of a standard is its stability in solution over time towards

ice nucleation. Therefore, we performed multiple FINC experiments of the same 20 mg C L$^{-1}$ solution of lignin (using Batch

2) over four months and found no change in ice nucleating activity (Fig. 7), unlike for NX-illite (see Sect. 5.2 and Fig. 4). We additionally performed several freezing experiments on a more concentrated solution of 200 mg C L$^{-1}$ lignin (using Batch 2), and also found no significant change in ice nucleating activity over eight weeks (Fig. S16). Furthermore, we tested the effect of the storage temperature of the lignin solution: in the fridge or on the lab bench and found no difference (Fig. S17). This solution stability makes lignin a complimentary ice nucleation standard over NX-illite and Snomax.

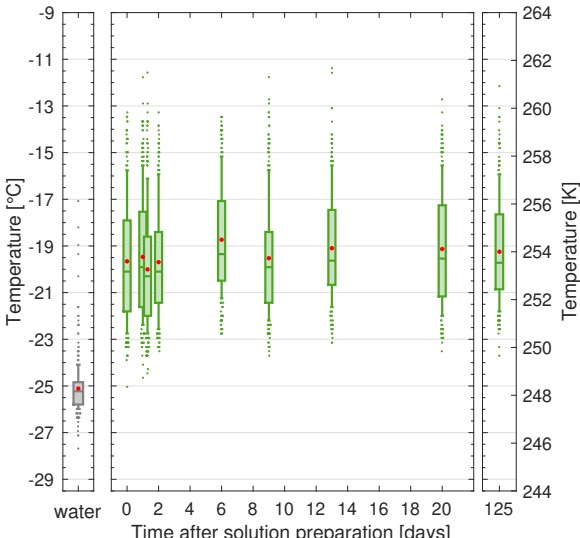

**Figure 7.** The solution stability of a 20 mg C L$^{-1}$ lignin solution (Batch 2) over 125 days, stored in the fridge in the dark. Each boxplot represents the frozen temperatures of 288 wells, where the middle line is the median ($T_{50}$), the outer edges of the box are the 25th and 75th percentiles, the whiskers extend to the 10th and 90th percentiles, and the red dot is the mean. The water boxplot represented the background water solution. Duplicates were taken on Day 1 (overlapping boxplots). All temperature measurements have an uncertainty of $\pm$ 0.5 °C.

## 6.5 Lignin's macromolecular size and reactivity

Lignin's recalcitrance towards atmospheric processing has recently been demonstrated by Bogler and Borduas-Dedekind (2020). Indeed, photochemical exposure, ozonation, heating, sonication, and hydrogen peroxide treatments had negligible effects on lignin's ice nucleation ability. Note also that the lignin recommended herein is soluble; accordingly, it passed through a 0.2 μm filter without loss of ice activity, but lost some activity after filtering to 0.02 μm (Bogler and Borduas-Dedekind, 2020). Lignin could therefore be termed a nano-INP (O'Sullivan et al., 2015), as an INM (Pummer et al., 2015) and operationally defined as soluble (Borduas-Dedekind et al., 2019). The ability of lignin to be chemically recalcitrant further demonstrates its value as a standard for DFTs.

## 7 Conclusions

We describe herein our home-built drop Freezing Ice Nuclei Counter (FINC) to measure the ice nucleating ability of INPs and INMs in the immersion freezing mode. FINC is a complimentary instrument and part of a growing list of droplet freezing techniques (Table 1). Its capabilities include (1) measuring simultaneously 3 trays of 96 wells each, (2) supercooling to $T_{50} = -25.4 \pm 0.2$ with 5 μL droplets with a temperature uncertainty of $\pm 0.5$ °C, and (3) automated controls for running an experiment. We further explored eight different possible explanations to non-homogeneous freezing behaviour observed in FINC and other DFTs. As part of the development and validation of FINC, we intercompared measurements with NX-illite suspensions and an ambient aerosol sample collected in Zurich with other DFTs, specifically with DRINCZ and LINDA. Additionally, we present evidence for the use of soluble lignin as a reproducible, easy to use, and commercial intercomparison standard in future ice nucleation studies in immersion freezing. Indeed, we demonstrated that solutions of 20 mg C L$^{-1}$ measured on FINC, DRINCZ, and LINDA yielded a spread of $n_m$ values within a factor of 3 between −8 and −25 °C. We subsequently fitted a parameterization ($n_m = exp(-0.558 * T - 3.12)$) through our empirical data for further use in intercomparison and validation studies. Finally, we showed that lignin solutions are stable over several months and their ice nucleating activity is well reproduced across different batches of the same product, making lignin a competitive solution standard.

*Code and data availability.* Data sets for all data presented in figures in the text are deposited in the ETH Research Collection data repository at https://doi.org/10.3929/ethz-b-000438875. Necessary code for running experiments on FINC is also available upon request.

*Author contributions.* K.P.B. and N.B.D. designed the instrument. K.P.B. wrote the code to operate the instrument. A.J.M. validated the instrument. A.J.M., K.P.B, C.M., and J.W. acquired data during the intercomparisons; N.B.D., A.J.M. and K.P.B. wrote the manuscript with contributions from all authors.

*Competing interests.* The authors declare no competing interests.

*Acknowledgements.* We acknowledge the technical help of Marco Vecellio and Michael Rösch for advice, ordering hardware and machining structural instrument components. We also acknowledge the assistance of Max Aragon Cerecedes during the intercomparison experiment day. We thank Sophie Bogler for help with the tray washing experiments and Silvan Müller for help with the TOC analyzer. We are thankful to Paul DeMott and Tom Hill for the reviewing our compiled list of droplet freezing techniques.

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
