# Peer review of "Development of the drop Freezing Ice Nuclei Counter (FINC), intercomparison of droplet freezing techniques, and use of soluble lignin as an atmospheric ice nucleation standard"

_Atmospheric Measurement Techniques, 2020_

## Editor Comment (EC1) · Mingjin Tang (Editor) · 11 Nov 2020

As the handling editor of this manuscript, I have two comments which I kindly ask the authors to consider: 1) The table of content may not be necessary. 2) The list of peer-reviewed DFTs is very valuable, and its impact can be increased if the list is included in the manuscript.
* * *

---

## Referee Comment (RC1) · Anonymous Referee #1 · 5 Dec 2020

General comments:

This reviewer would support publication of this manuscript in AMT after minor (but seemingly necessary) revisions. Though the development/introduction of another DFT would not add an entirely new aspect in the research community, the reviewer advocates the authors for the fact that they carefully address all the details behind their new DFT and beyond. The amount of trivial details over the main manuscript and SI (clipped from one's thesis?) are somewhat bulky and cumbersome to read in general, but this reviewer still considers these as positive and beneficial information and contribution to

[Figure]

AMT.

Minor and technical comments:

*P3L71: The authors may consider adding Murray & Koop (2016) J. Chem. Phys. and/or Koop (2000) Nature as for homogeneous freezing reference(s). These papers nicely discusses on water volume dependent homogeneous freezing etc., which seems relevant to the concurrent study.

*P3L71-73: It would be nice if the authors can elaborate on how 'dominant' immersion freezing is in a bit more quantitative manner in the text. This fundamental information seems important since FINC is specifically developed to look into immersion only.

*P3L79: "to improve estimates of" –> "as well as improving overall understanding in"; there has been an ongoing discussion on the relationship between INP and ice crystal concentrations. At this stage, the discussion of 'whether chicken is first or egg is first' is not settled as it involves many aspects, such as aerosol dynamics, cloud macro-/micro-physics incl. secondary ice formation etc. In any case, the statement implying INP estimation to improve ice crystal estimation (regardless of intension) is seemingly misleading as no definite cause-effect answer is currently available. A simple modified statement separating INP from ice crystals (vice versa) may be the safest thing the authors may do. The reviewer suggest the authors to give some considerations at the least.

*P3L84-P485: How about water types, detectable T ranges, uncertainties etc.? The reviewer believes that there are many other variables to be considered in this statement.

*P4L98-99: The authors might want to briefly discuss advantages and disadvantages in detail – these information would be meaningful/useful to the reader.

*P4L102-103: The authors may include the investigable T range (and a summary of other limitations) of FINC in this statement. Carefully including caveats to the reader is as important as offering sales points in any technique papers, in the reviewer's opinion.

[Figure]

*P5L120: How 'abundant' in the atmosphere? The authors may provide brief but quantitative information here for the reader.

*P6L153: The reviewer strongly suggests replacing "unique..." with "updated feature of existing DFTs (Sect. S1)." At the end, FINC is one of DFTs and claiming the novelty of another DFT might not be a right approach to go with the concurrent paper. Simply reducing the tone (here and everywhere applicable for the similar context) should resolve the issue.

*P6L157: Briefly explain what the oven heating is for here. The reviewer is aware that it is discussed in the later section, but doing this may increase the overall readability for the future reader.

*P6L164: How crucial is it? Briefly explain/summarize what is discussed in David et al. It should be straightforward, and the readers would appreciate this complementary summary info appearing here rather than going through another paper themselves.

*Fig. 2: So each image # corresponds to incremental step of +0.2 dC? Then, for clarity to the future reader, the authors may consider introducing the temperature axis (on the top x-axis?). Also, where does this data/result come from? Which sample? The authors may want to briefly mention it in the caption or associated text.

*Sect. 2.4: The authors might want to briefly describe their general/specific suspension dilution methods (any systematic procedures; e.g., x10, x100 for all etc.?) as well as the data merging protocols – how they dealt with the overlapping n_INP (T) data – in this section somewhere.

*Sect. 3: Errors/uncertainties of DFTs are typically evaluated and expressed for both temperature and INP counts (or any ice nucleation efficiency metrics, such as n_m, n_s etc.). In the reviewer's opinion, the DFT uncertainty cannot be governed by a single variable of temperature. Please discuss the uncertainty involved in FINC for its INP counts at the least in this section. For instance, the authors can compute

binomial confidence interval errors on their INP counts (e.g., at 95% - see Eqn 3.21 of https://publikationen.bibliothek.kit.edu/1000076327). In addition, related to this point, the reviewer finds it a bit strange that Figs. 4 and 6 dot not show any error bars. The reviewer urges the authors to show error ranges in any visual presentations of IN results. The reviewer is certain that the reader would appreciate it, too.

*P10L251-253: Adding Murray & Koop (2016) J. Chem. Phys. besides O & Wood (2016) plus extending the discussion of previous findings on measurable droplet homogeneous freezing would be meaningful.

*P10L259-260: if the water volume is ∼60 micro L, then it becomes more like bulk water, correct? Or the authors have a particular reason calling it still as 'droplet'? Rephrasing it may be needed.

*P10L263-265: How do these measured $T\_50$ values compared to theory (e.g., CNT)? Please elaborate it here.

*Sect. 4.3.2: I agree that there would not be inverse Kelvin effect for the given dimension, but the contact angles and all other relevant properties, that may matter for freezing, change depending on the volume used, correct? Would the authors elaborate this point a bit further within this section?

*Sect. 4.4.1: Please state if/how the authors applied background freezing corrections somewhere within this sub-section. Looking at a non-negligible contribution of pure water droplet freezing at above -25 dC (Figs. S3, S5), the authors may have corrected the other data presented in this paper for this background contribution in some ways (or not? – then, why?)? Please clarify. This may be something already mentioned in the manuscript and overlooked by the reviewer, but having an independent sub-section for the background corrections and all that may make a good, informative section.

*Sect. 4.4.4: The reviewer sees the point of all the bubble discussion. But, this section sounds a bit speculative. More quantitative proofs of supporting FINC and DFTs would

have problem with bubbles? Perhaps, dealing with highly viscous solutions (e.g., high wt% of Snomax) tend to make micro bubbles (perhaps not so visible) and introduce high deviation and low reproducibility of the immersion IN spectra because of bubbles?

*Sect. 4.4.6: Keeping things in a laminar hood is a good practice in general. The reviewer wonders how the measurements prepared and carried out outside the hood would impact immersion compared to all the operation conducted in a hood. The quantitative answer incorporated in this section might strengthen the paper.

*Fig. 3: The reviewer wonders how this result compare to previously published work of the freezing depletion by solute; e.g., Whale et al. (2018) Chem. Sci. Please elaborate it in Sect. 4.6.

*Fig. 4: As mentioned earlier, it would be really nice to see x- and y-axis uncertainties in this sort of figures (even only on several representative data points). The same suggestion goes to Fig. 6 in P19.

*Sect. 6 onwards: The reviewer likes the idea of seeking a chemically inert & stable standard for DFTs. The tones of some words/phrases/sentences in this section seem too strong for given the context (e.g., P21L497-499 – including but not limited to). The reviewer suggests carefully re-phasing some parts to simply report what are observed/measured. The reviewer feels more comfortable accepting the proposed idea of suggesting lignin as one of potential DFT standards if the limitations are also offered, too (nothing could be perfect for now, correct?). No special procedure seems needed for the lignin suspension preparation other than suspending correct mass of lignin in pure water – is this right?

*P18L437: Justification of why $n_m$ is appropriate to use as IN efficiency metric should be addressed here. Do the authors assume absolutely all lignin components are soluble without any insoluble precipitates? Would that be really the case? The reviewer is asking this question because, in P18L450, the authors states that this parameterization is limited specifically to 20 mg C $L^{-1}$. Why so? If $n_m$ is truly applicable and reasonably

representing IN efficiency of lignin, the n_m(T) of different lignin mass concentrations should overlap each other and collapse into a single spectrum with a minimum n_m(T) deviation, correct? This is a similar question/concern raised in Sect. 2.4 (i.e., how to merge/stitch different dilution spectra).

*Fig. 6: The reviewer is not so sure if the IN efficiency deviation involved in lignin is notably better than that of illite NX. The reviewer is aware that different batches were examined for Sigma Aldrich lignin in this study. Careful word choice seems necessary here.

*Cont'd: The reviewer is also aware that the authors' intension is to attempt to designate lignin and a potential standard for DFTs. Regardless, how do DFT-lignin data compare to online IN chamber results for lignin (Steinke et al., 2020, ACP)? The future reader will appreciate to see this discussion somewhere in this section.

*P18L454-455: This sounds like lignin exists in maritime sources. The reviewer suggests rephrasing this sentence.

*S1: The reviewer finds the compiled summary list of currently existing DFTs valuable. Are the authors or whoever is in charge of this data depository intending to continuously update contents of the list in the future (, which would be tremendous effort for the IN research community)? If so, please keep and log update dates etc. for each version. It may be a good idea if the authors or whoever is in charge of this data depository can contact each instrument PI to check if there are any updates periodically to keep all information valid, accurate, and up to date all the time? By the way, WT-CRAFT deals with 3 uL droplets, not 6 – just so you know. There may be other info to be revised/updated.

*S2: 0.2 dC instead of 2 dC?

*Fig. S2: What sample is this? The authors may want to briefly describe it here.

*Fig. S3 or S5: It would be nice to see the homogeneous nucleation frozen fraction

curves from numerical modeling and their respective droplet volume at a cooling rate of FINC – e.g., Koop and Murray (2016) – in comparison to the FINC pure water data.

*P1L5: quantification –> estimation; this word choice seems better fitting for the given context.

*P1L21: in the research field of

*P3L65 & P4L112: ice nuclei –> ice-nucleating particles – Any reason why ice nuclei is used in these two locations rather than just suing INPs uniformly as done in the rest of the manuscript?

The reviewer enjoyed reading this paper. Hope some of suggestions/comments made here help the authors (and future readers).

---

## Referee Comment (RC2) · Anonymous Referee #2 · 19 Dec 2020

General comments:

Miller et al. developed a droplet freezing assay to quantify ice nucleating particles (INPs) in immersion freezing mode. 288 microliter droplets can be observed simultaneously while cooling down to approx. -25 °C with a temperature uncertainty of 0.5 °C. The authors extensively discussed possible sources of contamination and performed an intercomparison study with two other droplet freezing assays to validate the new instrument. Additionally, they tested and discussed the water-soluble biopolymer lignin as a suitable ice nucleation standard material. The development of a new droplet

freezing assay is not a substantial new concept for the ice nucleation community, but the authors extensively describe, test and discuss their new instrument and also many aspects beyond. This aspect along with the research for a good ice nucleation standard is highly valuable for the community. The manuscript is suitable for publication in AMT after the following comments have been addressed.

Specific comments:

Line 66: The high freezing temperature of -1 °C was not only reported for bacterial IN (e.g., Maki et al. 1974) but also for fungal IN (e.g., Richard et al. 1996). Please consider to include this aspect as well.

Line 69: Please include the following references: Felgitsch et al. 2018, Kunert et al. 2019, Pummer et al. 2015.

Line 98: Please add the following reference: Kunert et al. 2018.

Lines 98f: It would be nice if the authors could elaborate a bit more on advantages and disadvantages of the different methods.

Line 112: Snomax does not only consist of proteins from P. syringae. It is rather a preparation of freeze-dried, irradiated cells from P. syringae, which are non-viable and damaged after this procedure. It contains all parts of the bacterial cells including IN-active proteins. Please clarify this sentence.

Line 153: Also other droplet freezing assays can measure INP with high statistics using other types of PCR trays as for example the high-throughput droplet freezing assay TINA, which can be operated with two 384-well plates in one experiment. Please attenuate the term "unique feature".

Lines 165ff: How do you ensure that the warm ethanol, which you add to the cooling bath during an experiment, does not affect your cooling rate of 1 °C per min?

Line 194: The link to Fig. S1 is not clear here. If you want to keep it in this sentence,

you should include two arrows in the picture showing the direction top to bottom and left to right.

Lines 205ff, 310ff: You should consider to add an additional preparation step and spin down the prepared PCR trays before placing them into FINC to remove possible bubbles and ensure a comparable position of the droplet in each well.

Lines 206f: I would recommend to move the sentence "We found that these cleaning procedures. . ." to line 202 after ". . .for at least one hour." and continue with "Sample solutions were then prepared. . .".

Lines 239f, 258, 263f, 489: I would recommend to only use one decimal as your temperature uncertainty is 0.5 °C and you cannot be more precise than that.

Line 349: What is mg C L-1?

Lines 373ff, 402ff, 434ff: How did you manage that the three instruments all measured aliquots of the same suspension? Did you move all instruments to one lab or did you prepare the solution and transported aliquots to the different locations? Please elaborate.

Line 453: The parameterization of Wilson et al. 2015 is far off the results obtained in this study. What is the additional benefit of including it in Fig. 6?

Lines 461f: Have you also tried to test lignin from a different supplier? It would be nice for a universal standard not to be dependent on one company.

Line 467: Several research groups within the ice nucleation community work additionally with the initial freezing temperature. The requirements for a good ice nucleation standard should be as well to have reproducible initial freezing temperatures. If I see correctly in Figure 6 (the blue colors are difficult to distinguish), the initial freezing temperatures are in a range at least between -13 °C and – 15 °C. Please elaborate a bit more on this aspect also with regard to other substances such as fungal IN, which have a highly reproducible initial freezing temperature even after different treatments

(Fröhlich et al. 2015, Kunert et al. 2019).

Line 472: I agree that there is no trend over time. But I cannot agree that the T50 values are all within the temperature uncertainty of the instrument. If I compare the medians in Fig. 7, I estimate that the difference between day 1 and day 6 is about 1 °C. Please correct your statement. It would be nice to also discuss the applicability of the initial freezing temperature here. Is lignin as standard only useful with regard to the T50 value?

Line 496: Why -38 °C? Fig. 6 shows only data until -25 °C, which is the limit of detection for your instrument.

Fig. 4: The quality of the figure could be improved. The symbols seem to be very blurry, which makes it hard to see. Also the light grey color is very difficult to follow.

Fig. 6: The blue colors are impossible to distinguish. Please choose more colors or at least more different blue colors.

Section S1: Location of device for TINA is Max Planck Institute for Chemistry in Mainz. Please correct.

Section S2: Here, you state that the camera takes a picture every 2 °C. How can the script then record an image every 0.2 °C (see line 175 of the manuscript)?

Table S2: I would recommend to only use one decimal as your temperature uncertainty is 0.5 °C and you cannot be more precise than that.

Technical comments:

Line 205: "solution"

Line 206: a parenthesis is missing after 4.4.4.

Line 238: please add "droplets" after "5 uL"

Line 247: "shape"

Line 251: "in FINC" not "on FINC"

Line 256: a parenthesis is missing after Wang (2013)

Line 268: "than" not "that"

Fig. S8 caption: Please remove the dot in "the. Milli-Q"

Sec. S8: Parentheses are missing after Eq. S2 and Eq. S4.

Figs. S9, S11, S12, captions: Missing dots at the end of the last sentences.

References:

Felgitsch, L., Baloh, P., Burkart, J., Mayr, M., Momken, M. E., Seifried, T. M., Winkler, P., Schmale III, D. G., and Grothe, H.: Birch leaves and branches as a source of ice-nucleating macromolecules, Atmos. Chem. Phys., 18, 16063–16079, 2018.

FroÌĹhlich-Nowoisky, J., Hill, T. C. J., Pummer, B. G., Yordanova, P., Franc, G. D., and PoÌĹschl, U.: Ice nucleation activity in the widespread soil fungus Mortierella alpina, Biogeosciences, 12, 1057–1071, 2015.

Kunert, A. T., Lamneck, M., Helleis, F., PoÌĹschl, U., PoÌĹhlker, M. L., and FroÌĹhlich-Nowoisky, J.: Twin-plate Ice Nucleation Assay (TINA) with infrared detection for high-throughput droplet freezing experiments with biological ice nuclei in laboratory and field samples, Atmos. Meas. Tech., 11, 6327–6337, 2018.

Kunert, A. T., Pöhlker, M. L., Tang, K., Krevert, C. S., Wieder, C., Speth, K. R., Hanson, L. E., Morris, C. E., Schmale III, D. G., Pöschl, U., and Fröhlich-Nowoisky, J.: Macro-molecular fungal ice nuclei in Fusarium: effects of physical and chemical processing, Biogeosciences, 16, 4647–4659, 2019.

Maki, L. R., Galyan, E. L., Chang-Chien, M. Caldwell, D. R.: Ice nucleation induced by Pseudomonas syringae, Applied Microbiology, 28, 456-459, 1974.

Pummer, B. G., Budke, C., Augustin-Bauditz, S., Niedermeier, D., Felgitsch, L., Kampf,

C. J., Huber, R. G., Liedl, K. R., Loerting, T., Moschen, T., Schauperl, M., Tollinger, M., Morris, C. E., Wex, H., Grothe, H., Poìĺschl, U., Koop, T., and Froìĺhlich-Nowoisky, J.: Ice nucleation by water- soluble macromolecules, Atmos. Chem. Phys., 15, 4077–4091, 2015.

Richard, C., Martin, J. G., and Pouleur, S.: Ice nucleation activity identified in some phytopathogenic Fusarium species, Phytoprotection, 77, 83–92, 1996.

---

## Referee Comment (RC3) · Anonymous Referee #3 · 6 Jan 2021

The authors present the development of a drop Freezing Ice Nucleation Counter (FINC), a droplet freezing technique (DFT), for the quantification of INP and INM concentrations in the immersion freezing mode. The authors used an NX-illite suspension and an ambient aerosol sample for an intercomparison (INP) study and propose herein the use of a water-soluble biopolymer, lignin, as a suitable ice nucleating (INM) standard.

The manuscript is well-written and fits into the journal Atmospheric Measurement Techniques. The paper should be published after revisions.

[Figure]

Main Comments

In general, I appreciate the idea of defining a standard for INM. However, I have doubts that lignin is a very suitable standard. As discussed by the authors, lignin is a biopolymer with an undefined molecular composition. Therefore the molecular formula in figure 5 makes only little sense. Instead, a mass spectrum (as a Van-Krevelen-Diagram) of the sample might give more information and gives the reader the possibility to compare other lignin samples to your standard.

In fact also your comparison of batches investigated concerning ice active mass site density (nm) in figure 6, is not very convincing since the nm values vary by an order of magnitude. This is quite a lot in comparison to other standards like K-feldspar or aged Snomax.

Of course is it a good idea to use a commercial product of reproducible characteristic. However, a product from pulp and paper industries is not guaranty for a steady composition. A NIST standard, e.g. Lignin CAS Registry Number: 8068-05-1, might be better suited. You should discuss these arguments in the paper.

In this context, I appreciate very much your discussion of aging of lignin, which I found very convincing.

Minor Comments

Fig.1: Add size bars to fig.1b and fig.1d. The reader who hasn't seen the set-up in reality, otherwise cannot judge the dimensions.

You might consider a table with similar droplet freezing experiments (see e.g. Table 1 in Häusler et al. Atmosphere, 9, 140, https://doi.org/10.3390/atmos9040140, 2018) discussing the pros and cons.

---

## Author Comment (AC1) · 5 Feb 2021

**Response to the editor and reviewers for the paper "Development of the drop Freezing Ice Nuclei Counter (FINC), intercomparison of droplet freezing techniques, and use of soluble lignin as an atmospheric ice nucleation standard"**
**by A.J. Miller, K.P. Brennan, C. Mignani, J. Wieder, R.O. David, and N. Borduas-Dedekind**

We thank the reviewers for their comments on our paper. To guide the review process we have copied the reviewer comments in black text. Our responses are in regular blue font. We have responded to all the referee comments and made alterations to our paper (**in bold text**).
* * *
**Editor's Comments**

As the handling editor of this manuscript, I have two comments which I kindly ask the authors to consider:

1) The table of content may not be necessary.
While we acknowledge that a table of contents is not typical for a paper in AMT, we believe it would be useful for readers to navigate the many sections of our manuscript and to effectively find the information they are looking for. However, we are happy to follow the final decision of the editors to decide whether a table of contents can be included in an AMT paper.

2) The list of peer reviewed DFTs is very valuable, and its impact can be increased if the list is included in the manuscript.
Thank you for the suggestion. Upon further consideration of the editor's comment, we have decided to add the table of DFTs to the manuscript. We were able to condense the table by consolidating information and without losing information. We hope the table of DFTs in the manuscript, rather than in a supplemental file, can be of greater value for future readers. The PI of this manuscript also intends on curating an up-to-date open access list of DFTs in the near future.
* * *
**Anonymous Referee #1**

General comments:

[R1.1] This reviewer would support publication of this manuscript in AMT after minor (but seemingly necessary) revisions. Though the development/introduction of another DFT would not add an entirely new aspect in the research community, the reviewer advocates the authors for the fact that they carefully address all the details behind their new DFT and beyond. The amount of trivial details over the main manuscript and SI (clipped from one's thesis?) are somewhat bulky and cumbersome to read in general, but this reviewer still considers these as positive and beneficial information and contribution to AMT.
Thank you for the positive feedback!

Minor and technical comments:

[R1.2] *P3L71: The authors may consider adding Murray & Koop (2016) J. Chem. Phys. and/or Koop (2000) Nature as for homogeneous freezing reference(s). These papers nicely discusses on water volume dependent homogeneous freezing etc., which seems relevant to the concurrent study.

Thank you for the suggestion. (Koop and Murray, 2016) was added to line 71, and (Koop et al., 2000) has been added to the discussion of freezing-point depression by salt in Section 4.6.

[R1.3] *P3L71-73: It would be nice if the authors can elaborate on how 'dominant' immersion freezing is in a bit more quantitative manner in the text. This fundamental information seems important since FINC is specifically developed to look into immersion only.

Indeed, we can be more quantitative in our introduction. Field observations of ice formation at temperatures relevant for mixed-phase clouds (MPCs) have only been observed after the presence of liquid clouds (e.g. (de Boer et al., 2011; Westbrook and Illingworth, 2013)). This observation implies that the ice formation proceeded through immersion freezing. For example, De Boer et al. (2011) concluded that immersion freezing was the dominant process in their observations of stratiform MPCs. Additionally, Westbrook and Illingworth (2013) reported immersion freezing as the dominant mechanism for long-lived MPCs. Assigning a specific number for the fraction of freezing events that occur in MPCs due to immersion freezing is challenging. Nevertheless, Hoose et al., 2010b, 2010a; Kanji et al., 2017; Murray et al., 2012; Tobo, 2016 report that immersion freezing is the dominant freezing mechanism for dust and biological particles with Hoose et al., (2010, ERL) reporting a value of 84% for all heterogeneous freezing events occurring via the immersion mode.

To address the comment by the reviewer, we modified our discussion to "The immersion freezing mode dominates heterogeneous freezing in mixed-phase clouds **(Hoose et al., 2010, De Boer et al., 2011, Murray et al., 2012, Westbrook et al., 2013, Tobo, 2016, Kanji et al., 2017)** and occurs when an INP or an INM nucleates ice from within a supercooled water droplet (Storelvmo **et al.,** 2017, Vali et al., 2015). **For instance, Hoose et al. (2010) reported that more than 85% of all heterogeneous freezing events in their simulation occured via the immersion mode.**".

[R1.4] *P3L79: "to improve estimates of" –> "as well as improving overall understanding in"; there has been an ongoing discussion on the relationship between INP and ice crystal concentrations. At this stage, the discussion of 'whether chicken is first or egg is first' is not settled as it involves many aspects, such as aerosol dynamics, cloud macro-/micro-physics incl. secondary ice formation etc. In any case, the statement implying INP estimation to improve ice crystal estimation (regardless of intension) is seemingly misleading as no definite cause-effect answer is currently available. A simple modified statement separating INP from ice crystals (vice versa) may be the safest thing the authors may do. The reviewer suggest the authors to give some considerations at the least.

Thank you for the clarification and suggestion. Generally (i.e. in the absence of sedimenting ice crystals from above, e.g. seeder-feeder effect),) primary ice formation (e.g., from INPs) is necessary for secondary ice formation. We are aware that the exact relationship between INP number and ice crystal number is not straightforward. We have added the reference to (Murray et al., 2021).

Therefore, the highlighted sentence has been replaced to avoid over-simplifications (lines 81-83): "Thus, the ability to **predict** INP and INM concentrations can improve estimates of primary **and secondary** ice concentrations in mixed-phase clouds, and thus can help reduce uncertainties in weather and climate projections **(Murray et al., 2021).**"

[R1.5] *P3L84-P485: How about water types, detectable T ranges, uncertainties etc.? The reviewer believes that there are many other variables to be considered in this statement.
We agree with the reviewer and the sentence has been modified to the following: "Bench-top methods vary by **many variables, including** cooling method, droplet generation, droplet size, droplet number, freezing detection method**, detectable freezing temperature ranges, and measurement uncertainties**."

[R1.6] *P4L98-99: The authors might want to briefly discuss advantages and disadvantages in detail – these information would be meaningful/useful to the reader.
Good point. We have added the table of existing drop-freezing instruments to the manuscript (instead of having it as a supplemental file), and thus some advantages and disadvantages are now more accessible in the main text. We have modified the text in this section to explicitly point to these qualities: "Each bench-top immersion freezing method has its advantages and disadvantages, **which vary for the type of samples of interest.** Herein, we compiled a summary of multi-drop bench-top immersion freezing instruments used for atmospheric ice nucleation measurements **that have been published between 2000 and 2020, shown in Table 1. Included in the summary table is a brief description of the operation of each instrument, the water background using reported protocol, the average drop size, and the average number of droplets per experiment. Generally, large operating temperature ranges, low background freezing temperatures, and high number of drops per experiment are advantageous qualities."**

[R1.7] *P4L102-103: The authors may include the investigable T range (and a summary of other limitations) of FINC in this statement. Carefully including caveats to the reader is as important as offering sales points in any technique papers, in the reviewer's opinion.
We agree that caveats are important, and we have added the following lines to the paragraph (a continuation of the response above): "As these types of instruments are not yet commercial, we also built our own drop Freezing Ice Nuclei Counter (FINC) using a cooling bath and an optical detection method. **In comparison to the existing methods, FINC fits well within the range of operating parameters with drop sizes of 5 - 60 μL, 288 drops per experiment, an operating temperature range of 0 to -32 ˚C, and background freezing at -25 ˚C (Table 1).** The advantage of FINC over existing similar methods is its automation of the ethanol level, its use of

288 wells to increase statistics, and its improved code for well detection and for harmonizing the output data."

[R1.8] *P5L120: How 'abundant' in the atmosphere? The authors may provide brief but quantitative information here for the reader.
We agree with the reviewer that we could have been more precise. We have revised the sentence to reflect measured concentrations in the literature. The sentence now reads, "Furthermore, lignin and its oxidation products are **present** in the atmosphere, emitted **for example during agricultural harvesting and biomass burning. For example, typical plume concentrations of 149 ng/m-3 were observed in Houston, TX (Myers-Pigg et al., 2016; Shakya et al., 2011)"**

[R1.9] *P6L153: The reviewer strongly suggests replacing "unique. . ." with "updated feature of existing DFTs (Sect. S1)." At the end, FINC is one of DFTs and claiming the novelty of another DFT might not be a right approach to go with the concurrent paper. Simply reducing the tone (here and everywhere applicable for the similar context) should resolve the issue.
We agree with the reviewer that our tone warranted editing. We have thus changed the sentence to**: "**The use of Piko PCR trays in FINC is an **updated feature of existing DFTs (Table 1**)." We have additionally gone through the text to reduce the tone of "overselling".

[R1.10] *P6L157: Briefly explain what the oven heating is for here. The reviewer is aware that it is discussed in the later section, but doing this may increase the overall readability for the future reader.
Good suggestion. To improve readability and clarity, we have added to Section 2.1.2 the reasoning for oven-heating the trays, as well as the evidence for it shown in Figure S3. The reference to this figure was originally (and mistakenly) made in Section 2.4. The sentence in Line 157 has been modified: "The trays are heated in an oven at 120 °C for at least **one** hour before use**; this procedure improves reproducibility of background water experiments (Fig. S3)**".

[R1.11] *P6L164:  How crucial is it?  Briefly explain/summarize what is discussed in David et  al. It should be straightforward, and the readers would appreciate this complementary summary info appearing here rather than going through another paper themselves.
Indeed the explanation is straightforward. We have modified the sentence to: "**T**o achieve reproducible measurements, **the ethanol level must submerge the** well throughout the **experiment to avoid the formation of vertical temperature gradients within the well** ((David et al., 2019))**."**

[R1.12] *Fig. 2: So each image # corresponds to incremental step of +0.2 dC? Then, for clarity to the future reader, the authors may consider introducing the temperature axis (on the top x-axis?). Also, where does this data/result come from? Which sample? The authors may want to briefly mention it in the caption or associated text.
The reviewer is correct. We have added a temperature (K) axis on the top x-axis of Figure 2, as suggested.The data now shown in Figure 2 is from a background water sample, which we have

added to the caption: "**The freezing experiment depicts the results of a background water measurement.**" An earlier line of the caption was also modified to account for the updated data: "(The first **90** images corresponding to an interval of **18** degrees are excluded here for simplicity)."

[R1.13] *Sect. 2.4: The authors might want to briefly describe their general/specific suspension dilution methods (any systematic procedures; e.g., x10, x100 for all etc.?) as well as the data merging protocols – how they dealt with the overlapping n_INP (T) data – in this section somewhere.
Two reviewers had similar questions and so we added a paragraph to section 2.4 to clarify our procedure. Briefly, we do not merge any dilutions series and all INP data is represented in our plots (every 288 points).

"**Note that our sample preparation procedure does not include dilution series; we make the lignin solutions with the required concentration from weighed solid lignin. We have previously proposed that lignin may be aggregating in solution, leading to concentration-dependant ice nucleation behaviour ((Bogler and Borduas-Dedekind, 2020)). Based on this hypothesis, we also do not conduct any data merging procedure for different concentrations or volumes (for example in Fig. S7). Furthermore, we do not subtract our background water values, and prefer to show all raw data in box plot formats (as in Fig. S6 and see also (Brennan et al., 2020)).**"

[R1.14] *Sect. 3: Errors/uncertainties of DFTs are typically evaluated and expressed for both temperature and INP counts (or any ice nucleation efficiency metrics, such as n_m, n_s etc.). In the reviewer's opinion, the DFT uncertainty cannot be governed by a single variable of temperature. Please discuss the uncertainty involved in FINC for its INP counts at the least in this section. For instance, the authors can compute binomial confidence interval errors on their INP counts (e.g., at 95% - see Eqn 3.21 of https://publikationen.bibliothek.kit.edu/1000076327). In addition, related to this point, the reviewer finds it a bit strange that Figs. 4 and 6 dot not show any error bars. The reviewer urges the authors to show error ranges in any visual presentations of IN results. The reviewer is certain that the reader would appreciate it, too.
We agree with the value of calculating uncertainties for INP counts as well as for temperature. To address this issue, we calculated the uncertainties for Fig 4a,b, and Fig 6 according to the equations in Supplemental Section **11**. Briefly, we do a propagation of error with the following variables:
- illite $n_s$: uncertainty in number of wells frozen, well volume uncertainty, illite weight concentration uncertainty, SA_BET uncertainty
- aerosol INP concentration: uncertainty in number of wells frozen, well volume uncertainty, impinger flow uncertainty, impinger water volume uncertainty
- lignin $n_m$: uncertainty in number of wells frozen, well volume uncertainty, lignin concentration uncertainty

The error bars are added in Figure 4a,b, and Figure 6 for only the $T_{50}$ values for one sample of each of the instruments to avoid over-crowding the data. The error bars include the temperature

uncertainty (x-direction) and the INP counts (y-direction). The errors in the INP concentrations are small however, and are not as obvious in the figures unfortunately. We also add a sentence to each appropriate area to point the reader to the calculations in the Supplemental and to describe the error bars in Fig. 4 and Fig. 6.

Lignin: "**Uncertainties in $n_m$ include a 1% error in TOC, 1% error in V_DRINCZ, 0.5% error in V_LINDA, 8% error in V_FINC (well volume errors are based on error in pipettes), and finally an error of +/-1 in the number of frozen wells. The uncertainty calculations are presented in Section S11 and are displayed in Fig. 6 as error bars on the T_50 of one sample from each instrument.**"

Illite: " **The uncertainties in $n_{s,BET}$ include a 1% error in SA_BET ((Broadley et al., 2012)), 1% error in C_illite, 1% error in V_DRINCZ, 0.5% error in V_LINDA, 8% error in V_FINC (well volume errors are based on error in pipettes), and an error of +/- 1 in the number of frozen wells. Uncertainty calculations are detailed in Section S11, and plotted in Fig. 4a as vertical error bars on the T_50 values for one sample from each instrument.** "

Ambient aerosol: "**The uncertainties associated with the INP concentrations are plotted in Fig. 4b as vertical error bars on the T_50 values for one sample from each instrument, and the calculations are described in Section S11. The uncertainties include 1% error in V_DRINCZ, 0.5% error in V_LINDA, 8% error in V_FINC (well volume errors are based on error in pipettes), an error of 1 in the number of frozen wells, and error in the Coriolis impinger sampling +/-10 L min$^{-1}$ error in flow rate, 0.5 mL error in sample volume).**"

[R1.15] *P10L251-253: Adding Murray & Koop (2016) J. Chem. Phys. besides O & Wood (2016) plus extending the discussion of previous findings on measurable droplet homogeneous freezing would be meaningful.
We thank the reviewer for the suggestion and have added the Murray and Koop (2016) reference. Each DFT has its own non-homogeneous behaviour and one characteristic of this behaviour is the detection limit measured by background water and covered in Table 1. We have added the following sentence to the text, "**DFTs tabulated in Table 1 which use drops in the microliter range also show this non-homogeneous freezing behaviour**."

[R1.16] *P10L259-260: if the water volume is 60 micro L, then it becomes more like bulk water, correct? Or the authors have a particular reason calling it still as 'droplet'? Rephrasing it may be needed.
We have changed the word "droplet" to "**drop**" for clarity. However, throughout the manuscript we interchangeably use droplets and drops to describe the volume of water held within a well. A strict volume (or size) cutoff at 60 microL between these two definitions does not exist to the best of our knowledge.

[R1.17] *P10L263-265: How do these measured T_50 values compared to theory (e.g., CNT)? Please elaborate it here.

We have attempted to answer the reviewer's comment in the following two points: (but we invite the reviewer to get in touch with us if we misunderstood the question)

1.  In section 4.1: Based on calculations that 50% of a droplet population of 5 μL-volume is predicted to freeze spontaneously (< 1 s) and thus homogeneously at -31.81 °C, whereas 60 μL-volume is predicted to freeze at -31.41 °C (equations from (Wang, 2013)), all the reported $T_{50}$ values are warmer than predicted by the CNT.

2.  We have attempted to calculate a theoretical frozen fraction for a population of 5 uL spherical droplets, following the assumption of spontaneous freezing occurring within 1 second used in Section 4.1. The theoretical frozen fraction has been added to Figure S5 with the accompanying text in Section S4: "**Additionally, a theoretical frozen fraction curve is shown, as predicted by classical nucleation theory (CNT) for a population of 5 μL spherical droplets. The coded script uses equations from (Wang et al., 2013). The equations allow us to calculate how fast a certain fraction of the population will freeze at any given temperature. Therefore, to calculate the theoretical frozen fraction curve, we calculated the temperature at which several fractions (0.00001, 0.05, 0.1, 0.15, 0.25, 0.5, 0.75, 0.85, 0.9, 0.95, 0.99999) became frozen homogeneously (< 1 second).**"

[R1.18] *Sect. 4.3.2: I agree that there would not be inverse Kelvin effect for the given dimension, but the contact angles and all other relevant properties, that may matter for freezing, change depending on the volume used, correct? Would the authors elaborate this point a bit further within this section?

In our PCR tray wells, the contact angles are the same for different volumes as the interface between the solution and the well wall remains the same. It may be worth adding here that our system uses solutions within a well, rather than a single drop deposited on a surface.
The contact angle between the surface of the PCR tray and the water/solution/suspension should be constant regardless of the amount of water/solution/suspension in the tray as it depends on the chemical interactions between the water/solution/suspension and the plastic the tray is composed of.

[R1.19] *Sect. 4.4.1: Please state if/how the authors applied background freezing corrections somewhere within this sub-section. Looking at a non-negligible contribution of pure water droplet freezing at above -25 dC (Figs. S3, S5), the authors may have corrected the other data presented in this paper for this background contribution in some ways (or not? – then, why?)? Please clarify.  This may be something already mentioned in  the manuscript and overlooked by the reviewer, but having an independent sub-section for the background corrections and all that may make a good, informative section.

There is no background correction of any kind in the data presented in this manuscript. In this study, our measurements are always warmer than the freezing temperatures of our non-homogeneous freezing background. There is therefore no need to make corrections. Nevertheless, we prefer to show both, the sample and the background water, frozen fractions as in Figure 7 and S15 (now added to the SI).

We added to the text**: "Finally, we add that no background corrections are made in our data analysis."**

We also added paragraph to Section 2.4 to further clarify our sample preparation and data analysis procedure: "**Note that our sample preparation procedure does not include dilution series; we make the lignin solutions with the required concentration from weighed solid lignin. We have previously proposed that lignin may be aggregating in solution, leading to concentration-dependant ice nucleation behaviour (Bogler et al. 2020). Based on this hypothesis, we also do not conduct any data merging procedure for different concentrations or volumes (for example in Fig. S7). Furthermore, we do not subtract our background water values, and prefer to show all raw data in box plot formats (as in Fig. S6 and see also Brennan et al. (2020)).**"

In addition, we have added a figure to the supplemental (Fig. S15) to show the background freezing of DRINCZ, LINDA, and FINC on the day of the intercomparison experiments. Each instrument has a different limit of detection (i.e. background water frozen fraction). All of our measurements of NX-illite, the aerosol sample, and lignin were warmer than the background freezing for all three instruments. We have added a mention of this in the following: "**For background measurements, purchased, bottled molecular biology-free reagent water (Sigma-Aldrich, W4502-1L) was used (background measurements reported in Fig. S15), and no background corrections were made.**"

[R1.20] *Sect. 4.4.4: The reviewer sees the point of all the bubble discussion. But, this section sounds a bit speculative. More quantitative proofs of supporting FINC and DFTs would have problem with bubbles? Perhaps, dealing with highly viscous solutions (e.g., high wt% of Snomax) tend to make micro bubbles (perhaps not so visible) and introduce high deviation and low reproducibility of the immersion IN spectra because of bubbles?

We agree with the reviewer that our discussion of bubbles contributing to non-homogeneous freezing in FINC may not be applicable to other DFTs. We hope that our discussion in Section 4.4.4 may raise flags for other researchers to consider this issue in their own system. The evidence presented in Fig. S12 is our attempt to prove the role of bubbles. We certainly agree that more discussion and tests would be useful, but for now lay outside the scope of this paper.

[R1.21] *Sect. 4.4.6: Keeping things in a laminar hood is a good practice in general. The reviewer wonders how the measurements prepared and carried out outside the hood would impact immersion compared to all the operations conducted in a hood. The quantitative answer incorporated in this section might strengthen the paper.

We have opted to lay on the cautionary side and work with the laminar flow hood for all our samples. A combination of "as clean as possible" approaches have led us to the sample preparation protocol outlined in Section 2.4. Nonetheless, we acknowledge that DRINCZ protocols do not currently include sample preparation in a laminar flow hood. Further discussions on contaminations are also included in Barry et al., 2021 and Polen et al., 2018.

[R1.22] *Fig. 3: The reviewer wonders how this result compare to previously published work of the freezing depletion by solute; e.g., Whale et al. (2018) Chem. Sci. Please elaborate it in Sect. 4.6.

The Whale et al., 2018 paper worked with concentrations of 0.15 M and 0.015 M to remove the issue of freezing point depression. The concentrations we used are 0 M, 1M, 2M, and 3 M to specifically observe the freezing point depression. We don't think the Whale et al. (2018) paper is particularly relevant to our discussion since the authors were investigating the role of solutes outside conditions of freezing point depression (or colligative melting point depression).

[R1.23] *Fig. 4: As mentioned earlier, it would be really nice to see x- and y-axis uncertainties in this sort of figures (even only on several representative data points). The same suggestion goes to Fig. 6 in P19.

We have added error bars to the $T_{50}$ values for one sample from each instrument to represent uncertainties. We calculated the uncertainties for Fig. 4a,b, and Fig. 6 according to the equations in Supplemental Section **11**. Briefly, we do a propagation of error with the following variables:

- illite $n_s$: uncertainty in number of wells frozen, well volume uncertainty, illite weight concentration uncertainty, SA_BET uncertainty
- aerosol INP concentration: uncertainty in number of wells frozen, well volume uncertainty, impinger flow uncertainty, impinger water volume uncertainty
- lignin $n_m$: uncertainty in number of wells frozen, well volume uncertainty, lignin concentration uncertainty

The error bars are added in Fig 4a,b, and Fig. 6 for only the $T_{50}$ values for one sample of each of the instruments to avoid over-crowding the data. The error bars include the temperature uncertainty (x-direction) and the INP counts (y-direction). The errors in the INP concentrations are very small however, and are not obvious in the figures. We also add a sentence to each appropriate area to point the reader to the calculations in the Supplemental and to describe the error bars in Fig. 4 and Fig. 6.

[R1.24] *Sect. 6 onwards: The reviewer likes the idea of seeking a chemically inert & stable standard for DFTs. The tones of some words/phrases/sentences in this section seem too strong for given the context (e.g., P21L497-499 – including but not limited to). The reviewer suggests carefully re-phasing some parts to simply report what are observed/measured.

We agree with the reviewer and have modified a number (~5 changes) of adjectives to tone down any perceived over-selling.

The reviewer feels more comfortable accepting the proposed idea of suggesting lignin as one of potential DFT standards if the limitations are also offered, too (nothing could be perfect for now, correct? Correct. No special procedure seems needed for the lignin suspension preparation other than suspending correct mass of lignin in pure water – is this right?

Correct. We can also add that lignin is dissolving in water to make a solution. It is not a suspension, which is why we argue that a soluble INM standard may have the additional advantage of reproducibility than an insoluble INM standard.

[R1.25] *P18L437: Justification of why n_m is appropriate to use as IN efficiency metric should be addressed here. Do the authors assume absolutely all lignin components are soluble without any insoluble precipitates?
Yes, and its definition that filtering the solution and measuring the freezing temperatures of the filtrate gives identical measurements (See Bogler and Borduas-Dedekind (2020)).

Would that be really the case? Yes. The reviewer is asking this question because, in P18L450, the authors states that this parameterization is limited specifically to 20 mg C L^-1. Why so?
Because in a previous study, we quantified lignin's concentration-dependent ice nucleating ability. We hypothesize that lignin forms aggregates in solution and therefore we stress that 20 mg C L$^{-1}$ is our standard solution (see text lines 476-485).

If n_m is truly applicable and reasonably representing IN efficiency of lignin, the n_m(T) of different lignin mass concentrations should overlap each other and collapse into a single spectrum with a minimum n_m(T) deviation, correct?
Correct, but it does not. In fact, many INMs appear to have this behaviour (on-going research in our group)

This is a similar question/concern raised in Sect. 2.4 (i.e., how to merge/stitch different dilution spectra).
We do not merge the data in any way. All lignin solutions were prepared independently and added to Fig. 6 without further data manipulation.

[R1.26] *Fig. 6: The reviewer is not so sure if the IN efficiency deviation involved in lignin is notably better than that of illite NX. The reviewer is aware that different batches were examined for Sigma Aldrich lignin in this study. Careful word choice seems necessary here.
The efficiency and reproducibility of lignin as a standard is an important part of our manuscript, and since it wasn't clear to the reviewer, we've made changes to our discussion to further clarify. To help, we have quantified the spread in n_m values explicitly in the text:

""The freezing temperatures of 20 mg C L$^{-1}$ compared well between FINC, DRINCZ, and LINDA, with overlapping n$_m$ traces, falling within **a factor of 3** (Fig. 6). **Specifically, at -14.8 ˚C, lignin n$_m$ values from the intercomparison range from 77 mg$^{-1}$ to 223 mg$^{-1}$, and at -19.5 ˚C the range is from 1760 to 4560 mg$^{-1}$. This spread is an improvement relative to NX-illite, which has ns values that span an order of magnitude (Fig. 4a).**"

[Figure]

Above is a figure for NX-illite from David et al. (2019) showing a spread larger than 1 order of magnitude, and larger than the spread we observed in our intercomparison of NX-illite (Fig. 4a). As for lignin, the intercomparison data between FINC, DRINCZ, and LINDA is indeed even less than an order of magnitude - the variance is within a factor of 3. The batch comparison does have a larger range, but this is still within an order of magnitude. We've added to the text a statement quantifying the spread of lignin values from the intercomparison, in Section 6.2.

[R1.27] *Cont'd: The reviewer is also aware that the authors' intention is to attempt to designate lignin and a potential standard for DFTs. Regardless, how do DFT-lignin data compare to online IN chamber results for lignin (Steinke et al., 2020, ACP)? The future reader will appreciate to see this discussion somewhere in this section.

The reviewer raises an interesting yet difficult comparison. The AIDA chamber used in the (Steinke et al., 2020) paper is an expansion experiment where aerosol particles within the cloud chamber first activate as water droplets and may subsequently freeze. Therefore, the particles need to first be solid to be able to be dispersed in the AIDA chamber. Further, our lignin results are difficult to compare because we report $n_m$ values and the Steinke paper reports $n_s$ values. In fact, our colleague Cyril Brunner at ETH Zurich attempted to disperse lignin powder in AIDA back in the spring of 2019, but he reported being unsuccessful since the lignin powder is too fine to be dispersed in AIDA (personal communication). On the other hand, Steinke et al. report size distributions of lignin (Figure S3 in their ACP paper) using a bettisizer. Was lignin in a solution or dry dispersed using a rotary brush? We are not sure. An idea could be to first aerosolize a lignin solution and then to dry the wet polydispersed aerosols. Nevertheless, we opt to keep our discussion in this paper solely to DFTs, but recognize that lignin could be tested in the future for online measurement techniques.

[R1.28] *P18L454-455: This sounds like lignin exists in maritime sources. The reviewer suggests rephrasing this sentence.

We have revised the sentence to say, "**For reference to other INMs**, the n_m parameterization by (Wilson et al., 2015) of sea-surface microlayer organic matter is included (Fig. 6)."

[R1.29] *S1: The reviewer finds the compiled summary list of currently existing DFTs valuable. Are the authors or whoever is in charge of this data depository intending to continuously update contents of the list in the future (, which would be tremendous effort for the IN research community)? If so, please keep and log update dates etc. for each version. It may be a good idea if the authors or whoever is in charge of this data depository can contact each instrument PI to check if there are any updates periodically to keep all information valid, accurate, and up to date all the time? By the way, WT-CRAFT deals with 3 uL droplets, not 6 – just so you know. There may be other info to be revised/updated.
We thank the reviewer for their interest in our summary list of DFTs (and thank you for the correction for WT-CRAFT)! As per the editor's suggestion, we have added the summary list to the manuscript directly, to make it more visible and valuable. We agree that it would be of great value to keep this list updated and public. In the near future, Nadine Borduas-Dedekind is planning to keep an updated list on a more easily accessible and update-able platform (e.g. github). She will additionally publicize it within the community to gain others' input to keep it accurate and up-to-date.

[R1.30] *S2: 0.2 dC instead of 2 dC?
Yes, fixed: "...taken every **0.2** °C".

[R1.31] *Fig. S2: What sample is this? The authors may want to briefly describe it here.
The example shown in Fig. S2 was of a background water experiment using 20 µL well volume. This information is added to the caption of Fig. S2: "Map of freezing temperatures for each well **in a freezing experiment of background water using 20 µL well volumes.**"

[R1.32] *Fig. S3 or S5: It would be nice to see the homogeneous nucleation frozen fraction curves from numerical modeling and their respective droplet volume at a cooling rate of FINC – e.g., Koop and Murray (2016) – in comparison to the FINC pure water data.
We appreciate the suggestion. We have attempted to calculate a theoretical frozen fraction for a population of 5 µL spherical droplets, following the assumption of spontaneous freezing occurring within 1 second used in Section 4.1. The theoretical frozen fraction has been added to Figure S5 with the accompanying text in Section S4: "**Additionally, a theoretical frozen fraction curve is shown, as predicted by classical nucleation theory (CNT) for a population of 5 µL spherical droplets. The equations from Wang et al. (2013) allow us to calculate how fast a certain fraction of the population will freeze at any given temperature. Therefore, to calculate the theoretical frozen fraction curve, we calculated the temperature at which several fractions (0.00001, 0.05, 0.1, 0.15, 0.25, 0.5, 0.75, 0.85, 0.9, 0.95, 0.99999) became frozen spontaneously (< 1 second).**"

[R1.33] *P1L5: quantification –> estimation; this word choice seems better fitting for the given context.

The change has been made**:** "[...] for the **estimation** of INP and INM concentrations in the immersion freezing mode."

[R1.34] *P1L21: in the research field of
The text is modified to: "in the **research** field of atmospheric ice nucleation."

[R1.35] *P3L65 & P4L112: ice nuclei –> ice-nucleating particles – Any reason why ice nuclei is used in these two locations rather than just using INPs uniformly as done in the rest of the manuscript?
We acknowledge that we do not have a substantial reason for using the two different terms - in our view the terms are interchangeable. However, we agree that consistency is important and we have thus changed "ice nuclei" to "ice nucleating particles" or "INPs".
In **Line 65:** "One important aerosol-cloud interaction is the ice nucleation of supercooled liquid water droplets caused by **ice nucleating particles (INPs).**"
In **Line 67:** "[...] and for other currently unidentified warm **INPs** (Lloyd et al., 2020)."
In **Line 122:** "Snomax has additionally been used as a bacterial **ice nucleating** standard[...]"

The reviewer enjoyed reading this paper. Hope some of suggestions/comments made here help the authors (and future readers)
We really appreciate this reviewer's constructive critical feedback. Thank you!
* * *
**Anonymous Referee #2**

General comments:

[R2.1] Miller et al. developed a droplet freezing assay to quantify ice nucleating particles (INPs) in immersion freezing mode. 288 microliter droplets can be observed simulta- neously while cooling down to approx. -25 °C with a temperature uncertainty of 0.5 °C. The authors extensively discussed possible sources of contamination and performed an intercomparison study with two other droplet freezing assays to validate the new instrument. Additionally, they tested and discussed the water-soluble biopolymer lignin as a suitable ice nucleation standard material. The development of a new droplet freezing assay is not a substantial new concept for the ice nucleation community, but the authors extensively describe, test and discuss their new instrument and also many aspects beyond. This aspect along with the research for a good ice nucleation stan- dard is highly valuable for the community. The manuscript is suitable for publication in AMT after the following comments have been addressed.

Specific comments:

[R2.2] Line 66: The high freezing temperature of -1 °C was not only reported for bacterial IN (e.g., Maki et al. 1974) but also for fungal IN (e.g., Richard et al. 1996). Please consider to include this aspect as well.

We agree with the suggestion, and the sentence has been modified to include this reference: "Heterogeneous freezing can occur at temperatures as warm as -1 °C for **certain bacterial (e.g., _P. syringae_;** (Morris et al., 2004)**) and fungal (e.g., _Fusarium_ species;** (Richard et al., 1996)**)** INPs as well as for other currently unidentified warm INPs ((Lloyd et al., 2020))."

[R2.3] Line 69: Please include the following references: Felgitsch et al. 2018, Kunert et al. 2019, Pummer et al. 2015.

The references were added, with the addition of the qualifier "e.g," prior to the references to make clear that this is not an exhaustive, all-inclusive list of studies which have reported ice nucleating macromolecules: "**[...] ice nucleating macromolecules (INMs) are also capable of freezing supercooled water droplets (e.g., (Felgitsch et al., 2018; Kunert et al., 2019; Pummer et al., 2012, 2015)**"

[R2.4] Line 98: Please add the following reference: Kunert et al. 2018.

Thank you, it's added: "[...] or with infrared detection (e.g., Zaragotas et al., 2016; Harrison et al., 2018**; Kunert et al., 2018.**"

[R2.5] Lines 98f: It would be nice if the authors could elaborate a bit more on advantages and disadvantages of the different methods.

In accordance also with Reviewer 1's comment R1.6, we have added further discussion on the existing drop-freezing experiments. We have also added the tabulated summary of all instruments to the main text (instead of as a supplemental file).

The text now reads: "Each bench-top immersion freezing method has its advantages and disadvantages, **which vary for the type of samples of interest.** Herein, we compiled a summary of multi-drop bench-top immersion freezing instruments used for atmospheric ice nucleation measurements **that have been published between 2000 to 2020, shown in Table 1. Included in the summary table is a brief description of the operation of each instrument, the water background using reported protocol, the average drop size, and the average number of droplets per experiment. Generally, large operating ranges, low background freezing temperatures, and high number of drops per experiment are advantageous qualities."**

[R2.6] Line 112: Snomax does not only consist of proteins from P. syringae. It is rather a preparation of freeze-dried, irradiated cells from P. syringae, which are non-viable and damaged after this procedure. It contains all parts of the bacterial cells including IN- active proteins. Please clarify this sentence.

Thank you for this clarification. The text is modified: "Snomax has additionally been used as a bacterial **ice nucleating** standard and consists of **freeze-dried, irradiated cells from** *P. syringae* ((Wex et al., 2015))."

[R2.7] Line 153: Also other droplet freezing assays can measure INP with high statistics using other types of PCR trays as for example the high-throughput droplet freezing assay TINA, which can be operated with two 384-well plates in one experiment. Please attenuate the term "unique feature".

We agree with the reviewer that our tone may have been over-selling. We have thus changed the sentence**,** also in accordance with the suggestion by Reviewer 1 in R1.9**: "**The use of Piko PCR trays in FINC is an **updated** feature in our tabulated list of DFTs (**Table 1**)."

[R2.8] Lines 165ff: How do you ensure that the warm ethanol, which you add to the cooling bath during an experiment, does not affect your cooling rate of 1 ∘C per min?

The cooling rate is set and controlled by the Lauda bath at 1 °C/min and is thus not affected by the temperature of the added ethanol... Indeed, when warm ethanol is added, the cooling power is automatically increased. In addition, a few mL of warm ethanol to a bath of 10 L has a negligible impact on the overall bath temperature.

[R2.9] Line 194: The link to Fig. S1 is not clear here. If you want to keep it in this sentence, you should include two arrows in the picture showing the direction top to bottom and left to right.

To help clarify, we have added to the image in Fig. S1 arrows indicating top to bottom and left to right, as suggested. We have also added numbers to the wells in the top left corner to indicate the well positions in the sorted output vector of freezing temperatures. To the text we have added:

"**Freezing temperatures are output as a single-column vector, sorted by well position going top to bottom and left to right (Fig. S1).**"

Fig. S1 caption: "**The white arrows indicate the direction of the sorted freezing temperature values by well position, and the white numbers indicate the position of the well in the vector (only 1-4 and 17-20 are shown as an example, and the well positions continue in this trend).**"

[R2.10] Lines 205ff, 310ff: You should consider to add an additional preparation step and spin down the prepared PCR trays before placing them into FINC to remove possible bubbles and ensure a comparable position of the droplet in each well.

Unfortunately, we are not sure what a "spin down" preparation would involve. Perhaps a centrifuge? If so, the PCR trays cannot be added to a centrifuge in order to remove the bubbles as the shape of the PCR tray is not well suited for a centrifuge and furthermore the clean film on top of the PCR tray is likely not sealed enough to prevent leakage. The sample solution is always in the same position inside the well due to gravity and capillary effect. In this discussion, we were trying to make the point that our observation of bubbles could be leading to additional non-homogeneous freezing behaviour.

[R2.11] Lines 206f:  I would recommend to move the sentence "We found that these cleaning procedures. . ." to line  202  after ". . .for  at least  one  hour." and continue  with  "Sample solutions were then prepared. . .".

Good point. We have modified this section and the "PCR trays" section (2.1.2). The reference to Fig. S3 is intended in the context of pre-treating the PCR trays, not in relation to the cleaning of glassware, and thus it was moved to Section 2.1.2: "The trays are heated in an oven at 120 ˚C for at least one hour before use; **this procedure improves reproducibility of background water experiments (Fig. S3).**"

[R2.12] Lines 239f, 258, 263f, 489:  I would recommend to only use one decimal as your tem-perature uncertainty is 0.5 ◦C and you cannot be more precise than that.

We agree with the reviewer. We have modified the values to contain just one decimal as suggested: "[...] resulting in an LOD $T_{50}$ of **-25.4 ± 0.2** ˚C. We note that a value of ± 3 σ can also be used and would lead to a similar background $T_{50}$ of **-25.4 ± 0.4** ˚C (Fig. S5)." We have reduced other temperatures mentioned in the text to one decimal place as well.

[R2.13] Line 349: What is mg C L-1?

Mg C L$^{-1}$ is a concentration unit meaning milligrams of organic carbon per liter. We have modified the sentences to add clarity: "To further characterize FINC, we tested the ice nucleation activity of solutions of dissolved organic matter (DOM) **and of sodium chloride (NaCl; 31434, Sigma Aldrich). The DOM solutions were at concentrations of 20 mg carbon per liter (mg C L$^{-1}$) and were samples obtained from** Jericho Ditch, part of the Great Dismal Swamp in Suffolk, Virginia, USA (sample collection reported in (Borduas-Dedekind et al., 2019))".

[R2.14] Lines 373ff, 402ff, 434ff: How did you manage that the three instruments all measured aliquots of the same suspension?  Did you move all instruments to one lab or did you prepare the solution and transported aliquots to the different locations?  Please elaborate.

We prepared the samples for intercomparison including the solution in our lab at ETH in Zurich, next to FINC and DRINCZ. The samples were transported in a cooler (by train from Zurich to Basel) to LINDA the following day (i.e. measurement day), with overnight storage in the fridge. We've added the following to the text for clarification: "**All samples meant for intercomparison were prepared at ETH Zurich on July 10, 2019. On the following day, measurements at all three instruments were conducted. As LINDA was located in Basel, one batch of aliquots was transported by train in a cooler in the morning from the preparation location (in Zurich) to the measurement location (in Basel). For background measurements, purchased, bottled molecular biology-free reagent water (Sigma-Aldrich, W4502-1L) was used (background measurements reported in Figure S15), and no background corrections were made.**"

[R2.15] Line 453:  The parameterization of Wilson et al.  2015 is far off the results obtained in this study. What is the additional benefit of including it in Fig. 6?

The reviewer raises a fair point. We included the parameterization of Wilson et al. (2015) because it is, to our knowledge, the only parameterization of INPs using $n_m$ values. We thought it could be useful to show how lignin's $n_m$ values differ from other INMs, giving insight into how ice-active lignin is in comparison to the sea surface microlayer.

[R2.16] Lines 461f: Have you also tried to test lignin from a different supplier? It would be nice for a universal standard not to be dependent on one company.

It is certainly a good idea to test lignin from a different supplier, though we have not yet had the chance to do so. However, we do try to make it clear in our manuscript that we are proposing as a standard this specific product of low-sulfonate lignin from Sigma Aldrich, to avoid any confusion.

[R2.17] Line 467: Several research groups within the ice nucleation community work additionally with the initial freezing temperature. The requirements for a good ice nucleation standard should be as well to have reproducible initial freezing temperatures. If I see correctly in Figure 6 (the blue colors are difficult to distinguish), the initial freezing temperatures are in a range at least between -13 °C and – 15 °C. Please elaborate a bit more on this aspect also with regard to other substances such as fungal IN, which have a highly reproducible initial freezing temperature even after different treatments (Fröhlich et al. 2015, Kunert et al. 2019).

We thank the reviewer for this suggestion; it's a point we had not previously considered. We know from the Fröhlich-Nowoisky et al., (2015) and Kunert et al. (2019) papers that ice-active biological material is often highly reproducible, yet appears in low concentrations, and often is not present in each drop/well.

Additionally, we think that the variability in initial freezing temperature is something that continues to be observed. There could be rare configurations (bonding order, surface groups etc.) of lignin that could be more ice active than the majority of the lignin configurations observed. Not to mention, it is difficult to rule out potential contaminants in the sample received from the manufacturer that could be responsible for uncommon warmer freezing events. In general, the first 1 or 2 freezing events are not needed for a calibration standard as they can be sample specific and not necessary to probe the response of an instrument. In fact, it could be argued that a general calibration material would be one that has a shallow freezing slope which spans several degrees and is quite reproducible, omitting the first few rare INPs/INMs. This issue is also discussed in Polen et al. (2018), and in Barry et al. (2021). Nevertheless, we wanted to address the reviewer's comment, so we have added a discussion on T_5 values as representative of initial freezing temperatures.

We have added a line in the text to address these ideas:
**"In addition, $T_{50}$ and $T_5$ values were -17.5 +/- 0.9 and -12.4 +/- 1.3 °C, respectively. Unlike Kunert et al. (2018) who observed reproducible initial freezing temperatures with Snowmax, we observe lignin's freezing temperatures being reproducible starting at $T_5$ values (where 5% of wells are frozen)."**

[R2.18.1 ] Line 472: I agree that there is no trend over time. But I cannot agree that the T50 values are all within the temperature uncertainty of the instrument. If I compare the medians in Fig. 7, I estimate that the difference between day 1 and day 6 is about 1 ∘C. Please correct your statement.

We thank the reviewer for this observation, as it is indeed correct that the spread of $T_{50}$s extends to 1 degree, not 0.5 degree. We have simply removed the sentence, leaving only the conclusion that there is no trend over time, as this is the most relevant and important result.

[R2.18.2] Line 472 continued: It would be nice to also discuss the applicability of the initial freezing temperature here. Is lignin as standard only useful with regard to the T50 value?

The reviewer makes an interesting point. Following discussions among the authors, we would argue that an initial freezing temperature is most appropriate for biological samples such as Snowmax (for example in (Kunert et al., 2018)). However, lignin does not display reproducibly initial freezing temperatures compared to Snowmax. Therefore, we would refer the reviewer to our lignin parameterization equation to compare $n_m$ values across the temperature range of -8 °C to -25 °C (line 540).

To clarify, the following sentences were added to Section 6.2: **"In addition, $T_{50}$ and $T_5$ values were -17.5 +/- 0.9 and -12.4 +/- 1.3 °C, respectively. Unlike Kunert et al. (2018) who observed reproducible initial freezing temperatures with Snowmax, we observe lignin's freezing temperatures being reproducible starting at $T_5$ values (where 5% of wells are frozen)."**

[R2.19] Line 496: Why -38 °C? Fig. 6 shows only data until -25 °C, which is the limit of detection for your instrument.

We thank the reviewer for spotting this error.The number in line 496 has been changed to -25 ˚C.

[R2.20] Fig. 4: The quality of the figure could be improved. The symbols seem to be very blurry, which makes it hard to see. Also the light grey color is very difficult to follow.

We thank the reviewer for this suggestion. We have improved the graphs in Fig. 4a,b by decreasing the transparency of the data points (i.e. making them opaque) so that the edges are sharper.

[R2.21] Fig. 6: The blue colors are impossible to distinguish. Please choose more colors or at least more different blue colors.

We have chosen 6 different shades of blue for the lignin batch comparison in Fig. 6, which we hope improves the readability of the figure.

[R2.22] Section S1: Location of device for TINA is Max Planck Institute for Chemistry in Mainz. Please correct.

Thank you for the correction. Although we have removed  the location information in Table 1 while moving it from the supplemental material to the main text, we will take note of this update for our future curated list of DFTs.

[R2.23] Section S2:  Here, you state that the camera takes a picture every 2 ◦C. How can the script then record an image every 0.2 ◦C (see line 175 of the manuscript)?
We thank the reviewer for spotting this mistake! The camera records an image every 0.2 ˚C, so what was written in Section S2 was a mistake. The line has been corrected: "...taken every **0.2 ˚C**".

[R2.24] Table S2: I would recommend to only use one decimal as your temperature uncertainty is 0.5 ◦C and you cannot be more precise than that.
We agree and have modified all numbers in Table S2, as well as throughout the text, to only contain one decimal.

Technical comments:
We thank the reviewer for catching these details!

[R2.25] Line 205: "solution"
Modified to "We recommend pipetting the **solution** volume [...]"

[R2.26] Line 206: a parenthesis is missing after 4.4.4.
Modified to: "(see Section 4.4.4**).**"

[R2.27]  Line 238: please add "droplets" after "5 uL"
Modified to: "[...] experiments using 5 µL **droplets** [...]"

[R2.28] Line 247: "shape"
Modified to**:** "and of droplet **shape** [...]"

[R2.29] Line 251: "in FINC" not "on FINC"
Modified to**:** "[...] measured **in** FINC."

[R2.30] Line 256: a parenthesis is missing after Wang (2013)
Modified to**:** "(equations from Wang et al. (2013)**)."**

[R2.31] Line 268: "than" not "that"
Modified to**:** "[...] with a larger spread **than** all other volumes."

[R2.32] Fig. S8 caption: Please remove the dot in "the. Milli-Q"
Modified in Figure S8 caption: "[...] our lab's water from the Milli-Q [...]"

[R2.33] Sec. S8: Parentheses are missing after Eq. S2 and Eq. S4.
Modified to**:** "(Eq. S2**)**" and : "(Eq. S4**)**"

[R2.34] Figs. S9, S11, S12, captions: Missing dots at the end of the last sentences.
In captions of S9, S11, and S12, periods have been added to the end.